

# Learning to Think Like a Cartoon Captionist: Humor Understanding With Multimodal Reasoning Models

## Abstract

Humor remains one of the most elusive challenges for artificial intelligence, demanding models to integrate visual perception, cultural knowledge, and creative reasoning. The New Yorker Cartoon Caption Contest (NYCC) offers a uniquely structured testbed for this problem, pairing images with thousands of captions, expert curation, and large-scale audience judgments. Prior work largely reduces humor to black-box classification or preference prediction, overlooking the step-by-step reasoning processes employed by human captionists. We introduce a framework for teaching multimodal language models (MLLMs) to reason like professional captionists. Central to our approach are captionist reasoning traces that decompose humor into incongruity detection, resolution construction, and punchline evaluation. Models are first adapted through continual pretraining on humor-focused corpora, then trained with supervised fine-tuning on captionist-style traces, and finally aligned with humor judgments using reinforcement learning with grounded perceptual rewards and stylistic rewards. Across NYCC-derived matching and ranking tasks, our models significantly outperform strong multimodal baselines. Beyond accuracy, they generate explanations that align with expert strategies and audience preferences. These results highlight humor as a powerful frontier for multimodal reasoning and demonstrate that combining explicit reasoning supervision with preference alignment offers a scalable path toward computational humor.

## 1 Introduction

Humor is often regarded as one of the highest forms of human intelligence. It thrives on ambiguity, incongruity, and cultural reference, elements that continue to challenge even the most advanced large language models (LLMs) (Kazemi et al., 2025). As such, humor is not only an intriguing domain in its own right but also a demanding probe of whether AI systems can move beyond surface-level pattern recognition toward creative, context-sensitive reasoning.

Cartoon captioning provides a particularly rich case study. A single image sets up a premise; a caption delivers a twist that reframes the scene in an unexpected way. Success requires detecting visual incongruities, grounding them in cultural knowledge, and resolving them with wit. This incongruity–resolution dynamic, long studied in psychology, makes caption contests a formidable test of multimodal reasoning.

The *New Yorker Cartoon Caption Contest (NYCC)* offers a unique natural laboratory for studying humor. Each week, thousands of readers submit captions for a cartoon, editors select finalists, and the public votes on a winner. This process yields a rare alignment between visual input, linguistic creativity, expert judgment, and crowd preferences. Unlike

conventional vision–language benchmarks, NYCC emphasizes creativity, cultural grounding, and human reception, dimensions where current multimodal models struggle.

Existing systems trained on general-purpose corpora can describe images or generate plausible text but often miss the essence of what makes a cartoon funny. They may fail to identify the incongruity, overlook subtle cultural allusions, or default to generic punchlines. While recent work explores preference alignment or incongruity-aware prompting, most approaches still reduce humor to shallow matching rather than modeling the reasoning process behind successful captions. However, even the strongest proprietary models often succeed through scale, unseen data sources, or implicit exposure to humor-rich corpora, rather than explicit reasoning skills. Open-weight models, by contrast, lack the multimodal abstraction and cultural grounding required to analyze incongruity, evaluate competing humorous interpretations, or justify caption preferences. As a result, prompting alone cannot endow them with the structured reasoning processes used by expert captionists. This gap motivates the need for transparent, reproducible training pipelines that teach open models to reason about humor rather than merely pattern-match it.

In this paper, we propose a framework for teaching models to **think like cartoon captionists**. Drawing inspiration from expert strategies (Wood, 2024), we introduce *captionist reasoning traces*: structured, step-by-step explanations that identify incongruities, explore candidate resolutions, and evaluate punchlines. These traces serve as cognitive scaffolds, converting humor from a black-box prediction problem into an interpretable reasoning task. To further align models with human judgments, we combine this supervision with reinforcement learning, guided by grounded visual and stylistic rewards as well as crowd preferences. Unlike large closed-source systems whose humor competence may stem from scale or proprietary data, our goal is to develop an open and interpretable training framework that explicitly instills captionist-style reasoning in accessible multimodal models.

Our experiments show that this reasoning-first approach improves performance on NYCC matching and ranking benchmarks. More importantly, it produces explanations that better reflect expert reasoning and audience taste, demonstrating that structured supervision and preference alignment can close the gap between machine outputs and human humor.

**Contributions.**

- We present a multi-stage framework, comprised of continual pretraining, supervised reasoning-trace training, and reinforcement learning, that operationalizes captionist reasoning for humor understanding.
- We show that even with modest, curated supervision, explicit reasoning traces significantly improve both accuracy and interpretability on NYCC tasks.
- We design perceptual and stylistic rewards tailored to humor, demonstrating that aligning with human preferences extends multimodal reasoning into the creative domain.

Ultimately, our results highlight humor as a demanding but rewarding frontier for multimodal reasoning, and demonstrate a concrete path toward computational humor.

## 2 Related Work

### 2.1 Computational Humor

Humor has long been studied in psychology and linguistics, most prominently through incongruity–resolution theories (Suls, 1972; Attardo, 1994), which view humor as the violation of expectations followed by a surprising resolution. Script opposition (Attardo & Raskin, 1991), frame-shifting (Coulson & Kutas, 2001), and Theory of Mind accounts (Samson, 2012) further emphasize cognitive mechanisms involved in joke comprehension. Recent cognitive science work suggests that humor also leverages distributional expectations from language, with LLMs showing partial success in detecting one-liners but falling short of human performance (Trott et al., 2025).

Early computational humor systems relied on handcrafted features or rule-based templates for puns, wordplay, and short jokes (Ritchie, 2001; Mihalcea & Strapparava, 2006; Doogan

et al., 2017). Later, data-driven corpora such as large joke datasets (Hossain et al., 2019; 2020) and the Unfun corpus of humorous/non-humorous headlines (West & Horvitz, 2019) enabled supervised classifiers. Recent work has scaled this approach with LLM-based edits, creating paired humor/non-humor data (Horvitz et al., 2024). While these studies establish useful baselines, they generally reduce humor to shallow signals and lack the multimodal grounding required for cartoon captioning.

Other recent benchmarks extend humor beyond text, such as minimally contrastive funny vs. non-funny images in HumorDB (Jain et al., 2025) or associative humor generation in the Japanese Oogiri game (Zhong et al., 2024). YesBut (Hu et al., 2024) targets two-panel juxtaposition humor, while DeepEval (Yang et al., 2024) evaluates deeper visual semantics, including a small humor subset.

## 2.2 The New Yorker Cartoon Caption Contest

The New Yorker Cartoon Caption Contest (NYCC) as become a focal resource for humor research, offering a structured alignment between images, captions, expert curation, and crowd judgments. Hessel et al. (2023) introduced a benchmark based on NYCC with tasks including caption matching, ranking, and explanation generation. Subsequent works expanded the scope: Zhou et al. (2025) crafted harder ranking splits (10-vs-1000, 30-vs-300); others studied crowd-sourced humor preferences at scale (Zhang et al., 2024). Despite this progress, existing approaches often treat caption selection as black-box classification or pairwise preference modeling. None attempt to reconstruct the reasoning process of a professional captionist—an omission our work addresses by explicitly modeling step-by-step incongruity detection, resolution framing, and stylistic evaluation.

## 2.3 Multimodal Reasoning and Preference Alignment

A growing body of work explores how to equip multimodal LLMs with structured, step-by-step reasoning. Recent approaches introduce long-form reasoning traces distilled from stronger models, demonstrating that carefully designed prompts and staged supervision can reduce hallucination and consistently improve performance (Xu et al., 2025; Thawakar et al., 2025; Liu et al., 2025a; Liao et al., 2025). Collectively, these studies emphasizes the value of explicit reasoning traces in steering multimodal models toward more reliable, system-2–style behavior.

In parallel, reinforcement learning with (verifiable) grounded rewards has emerged as a complementary strategy for alignment. Methods such as Vision-R1 (Zhan et al., 2025), Visual-RFT (Liu et al., 2025b), and Perception-R1 (Xiao et al., 2025) define task-specific or perceptual rewards that strengthen visual grounding and mitigate spurious correlations. These advances highlight the critical rol of reward design in stabilizing multimodal reasoning and ensuring that generated chains of thought remain both accurate and perceptually grounded.

Our work bridges these lines by introducing captionist-inspired reasoning traces for cartoons and designing perceptual and stylistic rewards tailored to humor, thereby extending preference alignment into one of the most challenging creative domains.

## 3 Approach

The central inspiration for our approach is Wood's analysis of the New Yorker Cartoon Caption Contest Wood (2024). Wood shows that successful captionists do not simply produce a clever line; they follow a structured cognitive process that begins with detecting the visual incongruity, continues through exploring possible narrative resolutions, and culminates in selecting the punchline that best reframes the scene. Humor thus arises from a progression of perception → reinterpretation → resolution, rather than isolated wordplay. We therefore adopt captionist-style reasoning traces that guide models to follow the same sequence of analytic steps.

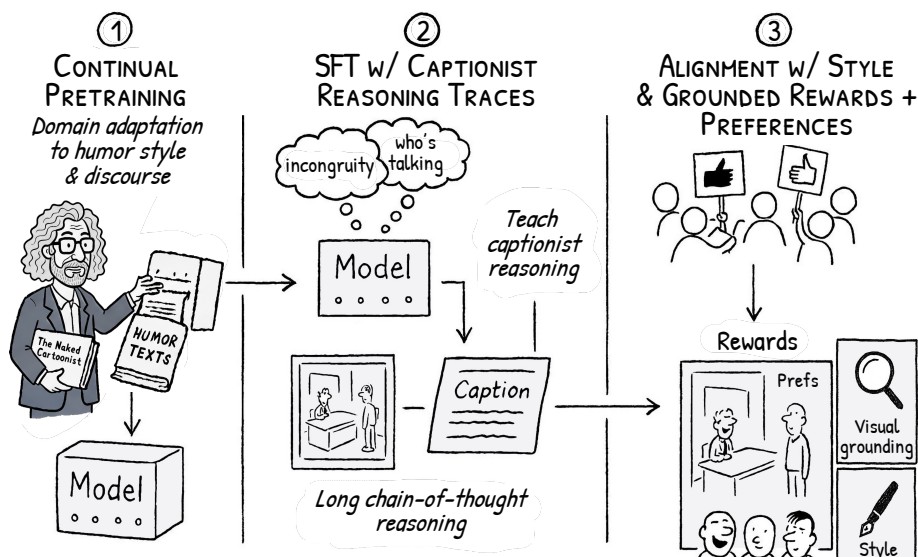

Figure 1: **Overview of our humor-alignment pipeline.** (1) *Continual pretraining* on humor-focused corpora adapts the backbone model to the discourse and stylistic conventions of cartoon humor. (2) *Supervised fine-tuning* on captionist reasoning traces teaches explicit step-by-step analysis of incongruity, resolution, and punchline fit. (3) *Reinforcement learning with grounded perceptual and stylistic rewards* further aligns model outputs with visual content, editorial tone, and crowd humor preferences using GPRO optimization.

Crucially, these traces are not generic LLM explanations. They encode the step-by-step analytic structure documented in captionist guides and editorial commentaries (Mankoff, 2002; 2014; Wood, 2024), including how to parse narrative roles, identify symbolic props, resolve incongruities, and apply culturally grounded heuristics. Because such traces implicitly assume familiarity with these interpretive conventions, we introduce a lightweight continual pretraining (CPT) stage to supply this background context. CPT does not aim to mimic a stylistic domain; rather, it provides the conceptual priors, drawn from judging discussions, captionist roundtables, and craft-oriented books, that allow supervised fine-tuning (SFT) to teach the structured reasoning procedure more effectively.

This design also parallels longstanding cognitive theories of humor, particularly incongruity–resolution frameworks, where humor results from detecting an expectation violation and arriving at a coherent reinterpretation. By supervising models with traces that explicitly instantiate this cognitive progression, our approach seeks to teach the underlying reasoning mechanism rather than relying on pattern matching.

To operationalize this, we adopt a three-stage framework (Figure 1). First, CPT provides relevant conceptual knowledge from humor-focused discussions and editorial analyses. Second, SFT uses structured prompt–response pairs to teach explicit captionist-style reasoning. Finally, reinforcement learning further aligns the model with perceptual grounding, stylistic conventions, and crowd preferences through targeted reward functions.

### 3.1 Continual Pretraining (CPT) for Humor Priors

Humor is culturally situated and stylistically constrained, yet standard pretraining corpora seldom capture the interpretive conventions involved in cartoon captioning. To supply this missing background, we perform **continual pretraining** on a curated corpus of humor-focused resources, including contest deliberations, editorial commentaries, caption-writing books, and complementary general-knowledge text.

This stage does not target downstream tasks directly. Instead, it acts as a **domain adapter**, biasing the backbone representation space toward humor-specific registers: incongruity–resolution structures, lexical economy, and cultural allusions. Training uses a standard causal

The scene in the cartoon shows an airplane cabin with passengers. The uncanny element is a giant amoeba or germ sitting in one of the seats, which is unusual because germs are microscopic. The man next to it is gesturing towards it, likely annoyed.
Looking at the captions:
A) "How'd you get yours off?" – This might relate to masks or something removable, but the amoeba isn't wearing anything. It doesn't fit.
B) "Mind if I cut through?" – This could imply someone trying to pass by, but the amoeba is blocking the aisle. However, the man is pointing at the germ, not asking to pass.
C) "Seven men walked into a bar and died. What's it to ya?" – This seems like a joke setup, but there's no bar in the scene. It's unrelated.
D) "I don't care if he's single-celled, he should have bought two seats." – This references the amoeba being single-celled (a biological term) and taking up space. The man is annoyed about the space, pointing at it. It fits perfectly.
E) " Unfortunately, the co-pay is an arm and a leg." – This is a medical cost joke, but there's no context for it in the cartoon. It doesn't fit.
The key points are the man's annoyance, the amoeba's size, and its encroachment on space. Caption D uses "single-celled" as wordplay, tying the biological aspect to the airline's seat policy. The humor comes from applying human rules to a microscopic organism that is now giant. The man speaking to the woman makes sense as he's complaining about the amoeba's lack of consideration. So D is the best match.

Figure 2: **Example captionist reasoning trace (matching).** Given one cartoon and five candidate captions (A–E), the trace reconstructs the visual setup, rules out off-topic options, and justifies the correct choice by linking the man's annoyance to the wordplay on "single-celled".

language modeling objective, and despite the modest dataset size, we observe that models fine-tuned after CPT exhibit more stable and coherent reasoning during SFT. *(Details and corpus composition are provided in Appendix A.)*

## 3.2 SUPERVISED FINE-TUNING (SFT)

To teach models how expert captionists reason, we construct a dataset of **reasoning traces** for both caption matching and ranking tasks. Starting from human-annotated descriptions of cartoons (Hessel et al., 2023), we generate structured step-by-step analyses using DeepSeek-R1 (DeepSeek-AI et al., 2025). These traces follow a fixed, human-derived reasoning template: scene reconstruction → incongruity identification → narrative resolution, rather than unconstrained LLM intuition. They cover scene reconstruction, incongruity identification, and humor justification.

We further *rephrase these traces with GPT-4o* (OpenAI, 2024) to make them resemble professional captionist commentary (e.g., concise, observational, and image-grounded). This rephrasing only adjusts style and fluency; it does not alter the underlying human-derived reasoning steps. In rare cases where annotations fail to support the correct caption, we lightly adjust the teacher prompt to steer the trace toward the right reasoning.

Formally, given an input pair $(I, Q)$ consisting of a cartoon image and its associated question, the model is fine-tuned on outputs $(y)$ formatted as:

$$\langle \texttt{think} \rangle \text{ reasoning steps } \langle /\texttt{think} \rangle \langle \texttt{answer} \rangle \text{ final choice } \langle /\texttt{answer} \rangle.$$

This **structured chain-of-thought supervision** encourages models to internalize captionist-style reasoning, beyond surface-level matching. *(See Appendix B for prompts and examples.)*

A representative trace is shown in Figure 2 where the model reconstructs the scene (giant amoeba on an airplane), rejects off-topic options, and chooses the caption that exploits the salient wordplay ("single-celled"). *(Additional examples are in Appendix D.)*

## 3.3 REINFORCEMENT LEARNING

While SFT equips models with reasoning patterns, outputs may still miss salient visual cues or stylistic aspects that shape humor. To further refine these behaviors, we apply **reinforce-**

**ment learning** using the GRPO algorithm DeepSeek-AI et al. (2025), which optimizes the reasoning process directly without a value network. Our reward design introduces targeted humor-related signals, such as grounding explanations in visual incongruities and adhering to captionist writing conventions, rather than just unconstrained humor preferences. The GRPO objective is:

$$
\mathcal{J}(\theta) = \mathbb{E}_{(I,q)\sim\mathcal{D},\ \{o_i\}_{i=1}^{G}\sim\pi_{\theta_{\mathrm{old}}}(O|I,q)}
$$

$$
\frac{1}{G}\sum_{i=1}^{G}\frac{1}{|o_i|}\sum_{t=1}^{|o_i|}\min\!\left(\frac{\pi_\theta(o_{i,t}\mid q,\,o_{i,<t})}{\pi_{\theta_{\mathrm{old}}}(o_{i,t}\mid q,\,o_{i,<t})}\,\hat{A}_{i,t},\,\mathrm{clip}\!\left(\frac{\pi_\theta(o_{i,t}\mid q,\,o_{i,<t})}{\pi_{\theta_{\mathrm{old}}}(o_{i,t}\mid q,\,o_{i,<t})},1-\epsilon,\,1+\epsilon\right)\hat{A}_{i,t}\right)
$$

$$
-\,\beta\,D_{\mathrm{KL}}\!\left[\pi_\theta\,\|\,\pi_{\mathrm{ref}}\right], \tag{1}
$$

where $\epsilon$ is the clipping hyper parameter, $\beta$ is the coefficient of the $D_{\mathrm{KL}}$, the KL-penalty term, and $\pi_{\mathrm{ref}}$ represents the reference model. The advantage $\hat{A}_i = \frac{r_i - mean(\{r_i\}_{i=1}^{G})}{std(\{r_i\}_{i=1}^{G}))}$ for the corresponding rewards $\{r_i\}_{i=1}^{G}$ of the responses $\{o_i\}_{i=1}^{G}$. This formulation allows us to directly optimize the interpretive reasoning process without relying on a scalar value baseline.

Our **composite reward**, involving four unique reward functions, integrates standard correctness checks with humor-specific signals, reflecting the dual nature of caption evaluation.

The first two are widely used in multimodal RLHF:

(i) **Accuracy ($R_a$)**, which verifies whether the final choice is correct: in matching, whether the selected caption is the gold caption; in ranking, whether the chosen caption matches the crowd-preferred one.

(ii) **Format ($R_f$)**, which ensures that responses follow the structured reasoning format `<think> reasoning </think> <answer> choice </answer>` for interpretability.

These provide reliable scaffolding but are not sufficient for deep humor reasoning. Our novelty lies in the following humor-oriented signals:

(iii) **Visual Perception ($R_p$)**, which rewards reasoning that explicitly grounds its explanation in salient objects, incongruities, and scene elements. To compute this reward, we first curated up to ten detailed descriptions per cartoon (entities, background, and incongruous objects), extended from Hessel et al. (2023). As illustrated in Figure 3, these curated references capture the salient visual details, which later serve as anchors for computing the perception reward. These anchors provide the reference set against which the Qwen2.5-7B-Instruct model (Qwen et al., 2025) acts as a judge, evaluating whether the model's descriptions are properly grounded in the visual details.

(iv) **Style ($R_s$)**, which evaluates captions from a linguistic perspective using an LLM-as-judge. Inspired by stylistic guidelines for caption writing (Wood, 2024), we constructed a prompt that evaluates five stylistic dimensions: use of natural everyday phrases, effective punctuation, creative wordplay, metaphorical expression, and placement of the punchline. Each dimension is scored in binary form, and the aggregated score constitutes the reward. For this reward, we employed Qwen-2.5-7B-Instruct model (Qwen et al., 2025) as well, encouraging the model not only to select the correct caption but also to justify it in a way that reflects the linguistic qualities of professional humor writing.

Figure 4 provides an example of how these judge-based rewards operate in practice: the perception judge verifies whether reasoning is visually grounded, while the style judge scores captionist features such as wordplay and punchline delivery. Their binary signals are aggregated into $R_p$ and $R_s$, directly guiding reinforcement learning toward visually faithful and editorially witty reasoning.

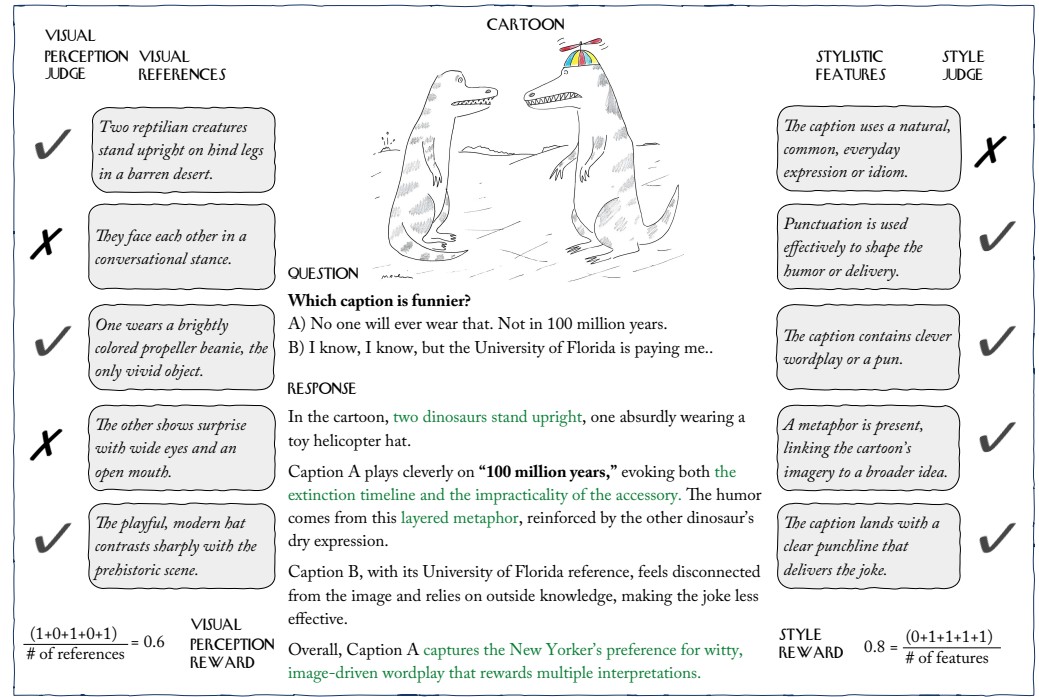

1. Numerous wire cages containing groups of small mice
2. Water bottles attached to the bars of the cages
3. A person wearing a full-body gray mouse costume with exaggerated ears, whiskers, and snout
4. The costumed person holding a clipboard and taking notes
5. A second person wearing a white lab coat
6. The lab-coated person standing slightly behind the costumed person
7. Real mice inside cages watching the costumed figure
8. The mouse costume is human-sized and plush in appearance
9. The costumed person positioned as if studying the mice in cages
10. Cages are stacked in the laboratory setting

Figure 3: **Curated visual references for a cartoon.** For each cartoon, we collect up to ten concise descriptions of entities, background, and incongruities; these serve as anchors for computing the perception reward.

Figure 4: **Example of judge-based rewards.** A cartoon with candidate captions evaluated by our two proposed judges. The *visual perception judge* checks whether the response grounds salient visual elements, while the *style judge* verifies linguistic qualities such as idiom, wordplay, metaphor, and punchline delivery. Binary scores are aggregated into the perception and style rewards ($R_p$, $R_s$), which shape reinforcement learning to encourage visually grounded and captionist-style reasoning.

The final reward is:

$$R = w_a R_a + w_f R_f + w_p R_p + w_s R_s, \tag{2}$$

Training on this reward signal teaches the model to not only answer correctly, but also to justify its choice in a way that reflects both **visual fidelity** and **editorial wit**. By removing the KL penalty (as in X-Reasoner (Liu et al., 2025a) and Perception-R1 (Xiao et al., 2025), we allow the policy to diverge from the reference model, improving its ability to generate coherent, captionist-style reasoning. *(Full prompts and judge templates are in Appendix C.)*

## 4 Experimental Setup

We evaluate our approach on four benchmark setups spanning caption matching and humor ranking. Two are drawn from the benchmark of Hessel et al. (2023) (matching and ranking tasks), and two are from Zhou et al. (2025), which contrast strong and weak captions at different rank gaps (10-vs-1000, 30-vs-300). Together, these cover both fine-grained discrimination and broad preference judgments.

### 4.1 Datasets and Tasks

Our evaluation covers four test setups spanning caption matching and humor ranking.

- **(Hessel et al., 2023):** (i) Matching, given an image and five candidate captions, select the gold caption; (ii) Ranking, given an image and two captions, select the one preferred by annotators.
- **(Zhou et al., 2025):** (iii) 10-vs-1000, discriminate between a top-10 caption and one ranked 1000–1009 by the crowd; (iv) 30-vs-300, discriminate between a caption ranked 30–39 and one ranked 300–309.

Unlike prior work, which relied on different splits, we curate four distinct evaluation settings to ensure robustness. To avoid leakage, all evaluation cartoons are removed from our continual pretraining corpus.

### 4.2 Models Tested

We evaluate our method against diverse set of models *(See Appendix C for the eval prompts)*:

- **Text-only reasoning model.** We include DeepSeek-R1 (DeepSeek-AI et al., 2025), a state-of-the-art reasoning LLM trained with reinforcement learning. Since it cannot process images directly, we evaluate it on ground-truth textual annotations of the cartoons (e.g., object/entity descriptions provided by Hessel et al., 2023). This setting gives it privileged access to curated scene details that multimodal models must infer from raw pixels, making its performance an upper bound for reasoning given perfect perception.
- **Closed multimodal models.** We evaluate two proprietary models with strong multimodal reasoning capabilities: o3, a reasoning-optimized variant of OpenAI's GPT-4 family, known for strong step-by-step inference; and o4-mini, a lighter multimodal model designed for efficient deployment. These systems represent the current frontier of proprietary multimodal LLMs.
- **Open multimodal reasoning models.** We further benchmark against recent open-weight models explicitly trained for multimodal reasoning: GLM-4.1V-9B-Thinking (Hong et al., 2025), which incorporates scalable reinforcement learning; Kimi-VL-A3B-Thinking-2506 (Team et al., 2025), which combines reasoning-augmented pretraining with preference optimization; LlamaV-o1 Thawakar et al. (2025) which gradually acquires skills for complex reasoning via multi-step curriculum learning, Qwen2.5-VL-7B-Instruct, and Qwen2.5-VL-32B-Instruct (Qwen et al., 2025), a stronger model that provides a more capable open baseline. These represent the latest openly-available multimodal reasoning LLMs.
- **Ours (7B backbone).** Our main model adapts the Qwen2.5-VL-7B-Instruct backbone with continual pretraining (CPT), supervised fine-tuning (SFT) using captionist reasoning traces, and reinforcement learning (RL) with perceptual and style rewards.
- **Ours (32B backbone).** To assess scalability, we additionally apply our full CPT–SFT–RL pipeline to the stronger Qwen2.5-VL-32B-Instruct backbone. This variant, denoted Ours-32B, achieves the best performance in our experiments and approaches or surpasses closed-source systems on our tasks.
- **Humans.** We report both expert and nonexpert baselines on a subset of the evaluation questions. The professional captionist serves as a qualitative anchor in that his preferences often favor originality over crowd-pleasing humor, while a user study with

Table 1: **Performance across models.** Accuracy (%) on caption matching and ranking tasks from Hessel et al. (2023) and Zhou et al. (2025). Bold values indicate the best performance in each column, and underlined values indicate the second-best performance (excluding human baselines).

| Model | (Hessel et al., 2023) | | (Zhou et al., 2025) | |
| --- | --- | --- | --- | --- |
| | Matching | Ranking | 10-vs-1000 | 30-vs-300 |
| Human (Expert) | 100.00 | 100.00 | 60.00 | 40.00 |
| Human (Non-Expert) | 53.03 | 65.61 | 54.70 | 52.27 |
| DeepSeek-R1 | 74.00 | 64.67 | 56.86 | 47.14 |
| o4-mini | 75.08 | 62.59 | 60.17 | 51.42 |
| o3 | **83.33** | 62.85 | **69.05** | **54.57** |
| GLM-4.1V-9B-Thinking | 59.40 | 55.60 | 52.28 | 49.14 |
| Kimi-VL-A3B-Thinking-2506 | 54.00 | 57.45 | 52.30 | 52.01 |
| LlamaV-o1 | 43.67 | 50.90 | 48.57 | 49.42 |
| Qwen2.5-VL-7B-Instruct | 42.67 | 55.06 | 50.57 | 47.99 |
| Qwen2.5-VL-32B-Instruct | 46.67 | 49.87 | 52.00 | 44.57 |
| Ours-7B | 60.33 | 62.08 | 53.14 | 49.43 |
| Ours-32B | 62.67 | **68.05** | 62.86 | 53.14 |

21 participants provides an additional measure of human–crowd alignment and helps contextualize model performance.

## 5 EXPERIMENTAL RESULTS

### 5.1 COMPARISON WITH BASELINES

Table 1 reports accuracy across all systems. As expected, DeepSeek-R1 performs strongly despite lacking visual inputs, since it operates on curated textual scene annotations and therefore serves as an *upper bound for reasoning given perfect perception* rather than a true multimodal comparison. The human expert scores perfectly on Matching and Ranking, tasks built from NYCC finalist captions, but aligns less with crowd preferences in the 10-vs-1000 and 30-vs-300 settings, consistent with the known divergence between editorial judgments and popular vote.

Among multimodal baselines, closed-source models (o3, o4-mini) remain the strongest on average. Their advantage may stem in part from large-scale pretraining on proprietary or paywalled content. Notably, OpenAI has entered a multi-year content licensing partnership with Condé Nast, the publisher of The New Yorker, which provides access to extensive archives of editorial material (ope, 2024). While this does not imply direct training on NYCC captions, it does suggest that closed models may benefit from stylistically similar, high-quality humor corpora unavailable to open-weight systems.

Open-weight models show a pronounced gap. Qwen2.5-VL-7B-Instruct significantly underperforms relative to closed systems, and even the larger Qwen2.5-VL-32B-Instruct only modestly improves performance, highlighting the difficulty of humor reasoning for general-purpose open models.

Our approach yields consistent and substantial improvements across both backbones. On the 7B model, the full CPT+SFT+RL pipeline markedly narrows the gap with proprietary multimodal systems, demonstrating that captionist-style reasoning can be effectively transferred even at small scale. When applied to the 32B backbone, the same pipeline produces the strongest open-weight model in our evaluation, outperforming all other open systems across all settings. Most notably, the 32B variant achieves the highest Ranking performance among all multimodal models we tested, surpassing even closed-source systems such as o3.

## 5.2 Ablation on Training Stages

Table 2 analyzes the contribution of each stage. Adding CPT before SFT improves adaptation but underperforms relative to SFT+RL. SFT is the main driver of gains, equipping the model with structured reasoning. RL alignment adds further improvements by enforcing perceptual grounding and stylistic constraints, with CPT+RL particularly strong on ranking. The full pipeline (CPT+SFT+RL) achieves the best overall results, showing that each stage contributes complementary benefits. Additional ablations are reported in Appendix G.

Table 2: **Ablation on model training stages.** Incrementally adding CPT, SFT, and RL yields complementary gains, with the full pipeline achieving the best performance.

| Approach | (Hessel et al., 2023) | | (Zhou et al., 2025) | |
| --- | --- | --- | --- | --- |
| | Matching | Ranking | 10-vs-1000 | 30-vs-300 |
| Base | 42.67 | 55.06 | 50.57 | 47.99 |
| Base + CPT + SFT | 49.00 | 56.88 | 54.57 | 49.14 |
| Base + SFT + RL | 57.33 | 58.18 | 56.29 | 44.86 |
| Base + CPT + RL | 46.00 | 51.95 | 49.43 | 50.29 |
| Base + CPT + SFT + RL (Ours) | 60.33 | 62.08 | 53.14 | 49.43 |

## 5.3 Ablation on Reward Functions

Table 3 isolates the contribution of different reward signals. Accuracy and format rewards alone provide considerable gains. Adding perceptual and style rewards provide useful, especially on the harder tasks from (Zhou et al., 2025), where referencing visual incongruities, and editorial tone and linguistic wit matter most. This shows that humor understanding requires both *what* is referenced (visual/perceptual detail) and *how* it is expressed (style).

Table 3: **Ablation on reward functions.** Humor-aware rewards substantially improve ranking and generalization.

| Approach | (Hessel et al., 2023) | | (Zhou et al., 2025) | |
| --- | --- | --- | --- | --- |
| | Matching | Ranking | 10-vs-1000 | 30-vs-300 |
| Base + CPT + SFT | 47.67 | 42.60 | 42.57 | 41.71 |
| + RL (w/ $R_a + R_f$) | 60.67 | 60.52 | 53.71 | 46.29 |
| + RL (w/ $R_a + R_f + R_p$) | 53.33 | 57.66 | 55.14 | 49.14 |
| + RL (w/ $R_a + R_f + R_s$) | 54.67 | 60.00 | 54.86 | 46.29 |

## 6 Conclusion

We proposed a three-stage training pipeline for multimodal reasoning about humor, using the New Yorker Cartoon Caption Contest as a challenging testbed. By combining humor-focused continual pretraining, supervised fine-tuning with captionist reasoning traces, and reinforcement learning with perceptual and stylistic rewards, our model learns not only to identify the correct caption but also to justify it with coherent, captionist-style reasoning.

Experiments across multiple evaluation splits demonstrate consistent improvements over general-purpose multimodal baselines, particularly in handling incongruity, visual grounding, and stylistic conventions. Ablation studies confirm the complementary roles of humor-specific pretraining and our novel reward design.

More broadly, this work shows that domain-adapted reasoning traces and targeted reward signals can enrich multimodal LLMs in tasks where subjective quality, stylistic nuance, and cultural grounding are essential. Future directions include scaling to larger backbones, exploring more efficient reward modeling, and extending the framework to other creative and subjective reasoning domains.

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

APPENDIX

We include this appendix to ensure full transparency and reproducibility of our work. It expands upon corpus construction, training configurations, prompt design, and qualitative analysis that could not be included in the main text due to space limits. The content is organized as follows:

- Section A–Continual Pretraining Corpora: Full documentation of our humor-focused CPT dataset, including composition, temporal coverage, and licensing information.
- Section B–Training Details: Model configurations, hyperparameters, optimization settings, and implementation notes for CPT, SFT, and RL stages.
- Section C–Prompt Templates: Complete prompt specifications used for generating reasoning traces, reward-judge evaluations, and model inference during testing.
- Section D–Qualitative Examples: Additional examples of model outputs at different training stages, including generated reasoning traces, RL-improved explanations, curated visual references, judge responses, and comparison with expert traces.
- Section E–Limitations and Ethical Considerations: Extended discussion expanding on failure modes, perception errors, humor subjectivity, cultural variability, and ethical concerns.
- Section F–Cross-Dataset Generalization: Additional experiments evaluating our model on external humor-related datasets, demonstrating that the reasoning patterns learned from NYCC transfer beyond the original domain.
- Section G–Additional Ablations on Training Stages: Comprehensive ablation experiments analyzing the individual and combined contributions of CPT, SFT, and RL to final model performance.

## A    Continual Pretraining Corpora

**Corpus Composition.**    Our continual pretraining (CPT) corpus was curated to capture the discourse and stylistic registers of cartoon humor. Table A.1 summarizes the main sources. These fall into three categories:

- **Contest deliberations & roundtables (audio/video)** supply conversational traces of caption brainstorming and judging: the *New Yorker Cartoon Caption Contest Podcast*, the *Official New Yorker Cartoon Podcast*, *CartoonStock YouTube Panel Discussions* (recorded judging sessions), and *The Cartoon Pad* (cartoonist roundtables).
- **Editorial & captionist commentary (written)** contributes reflective analyses that explain why finalist and winning captions work; we include Lawrence Wood's published commentaries on CartoonStock.com.
- **Books: craft, process, and editorial perspective** provide compact heuristics and creative guidance: *The Naked Cartoonist* (Robert Mankoff, 2002) (creativity/guide), *Your Caption Has Been Selected* (Lawrence Wood, 2024) (contest process and strategy), and *How About Never—Is Never Good for You?* (Bob Mankoff, 2014) (memoir with editorial insights).

As listed, to stabilize training and prevent overfitting, we further include subsets from FineWeb (Penedo et al., 2024) and OLMo-Mix-1124 (AllenAI, 2024), which provide general linguistic coverage and cultural references frequently invoked in caption writing. All items overlapping evaluation cartoons/captions are removed.

**Temporal Coverage.**    As summarized in Table A.2, our podcast and commentary sources span nearly a decade of caption contest discourse. The *Official New Yorker Cartoon Podcast* provides the earliest coverage (2015–2021), while more recent series such as the NYCC Podcast (2021–2024) and *The Cartoon Pad* (2021–2024) extend into the present. More recent commentary comes from CartoonStock YouTube Panels through July 2025, complemented by Lawrence Wood's weekly contest analyses (2019–2024).

Table A.1: **Composition of the continual pretraining corpus.** Instances, word counts, token counts, and share of total tokens by source.

| Source | Instances | Words | Tokens | % of Tokens |
|---|---|---|---|---|
| **Contest Deliberations and Commentaries** | | | | |
| New Yorker Caption Contest Podcast | 173 | 2,292,458 | 2,921,356 | 46.43% |
| The Cartoon Pad | 43 | 443,774 | 568,849 | 9.04% |
| Official New Yorker Cartoon Podcast | 34 | 349,588 | 465,701 | 7.40% |
| CartoonStock YouTube Panel Discussions | 36 | 224,864 | 281,295 | 4.47% |
| CartoonStock Lawrence Wood Commentaries | 175 | 128,635 | 170,978 | 2.72% |
| **Books on Caption Writing** | | | | |
| How About Never | 1 | 40,621 | 56,010 | 0.89% |
| Your Caption Has Been Selected | 94 | 8,087 | 11,057 | 0.18% |
| The Naked Cartoonist | 1 | 3,062 | 4,076 | 0.06% |
| **General-Purpose Corpora** | | | | |
| FineWeb | 1759 | 979,418 | 1,308,241 | 20.79% |
| Olmo-Mix-1124 | 1807 | 353,705 | 504,177 | 8.01% |
| **Total** | 4123 | 4,824,212 | 6,291,740 | 100.00% |

Table A.2: **Temporal coverage of humor-specific sources.** Podcasts and commentaries span nearly a decade of NYCC-related discussion, ensuring exposure to evolving audience expectations and cultural references.

| Source | First episode | Last episode | Coverage |
|---|---|---|---|
| NYCC Podcast | Mar 18, 2021 (Ep. 1) | Sep 25, 2024 (Ep. 173) | 3.5 years |
| The Cartoon Pad | Apr 2, 2021 (Ep. 1) | Jul 24, 2024 (Ep. 44) | 3 years |
| Official NYer Cartoon Podcast | Nov 14, 2015 (Ep. 29) | Jan 22, 2021 (Ep. 289) | 5+ years |
| Cartoon Stock YouTube Podcasts | Dec 20, 2022 | Jul 25, 2025 | 2.5 years |
| Lawrence Wood Commentaries | Jul 12, 2019 | Oct 22, 2024 | 5 years |

CartoonStock is an independent cartoon archive and marketplace that hosts a caption contest similar to NYCC. Its panel discussions and Wood's critiques are valuable because they explicitly analyze finalist and winning captions, highlighting what works and what fails. This breadth of coverage exposes the model to both long-term stylistic conventions and the evolution of humor preferences across audiences and platforms.

**Licensing note.** Parts of the CPT corpus are derived from copyrighted sources (e.g., published books, podcast transcripts). Due to licensing restrictions, we cannot release these directly. Instead, we will release the full train/test splits used for evaluation, and all prompts and code required to reproduce the preprocessing pipeline. This ensures reproducibility while respecting intellectual property constraints.

## B  TRAINING DETAILS

We describe training procedures for the three stages of our pipeline: continual pretraining (CPT), supervised fine-tuning (SFT), and reinforcement learning (RL). Table B.1 summarizes hyperparameters, while Figures B.1–B.3 illustrate training dynamics.

**Continual Pretraining (CPT)**  The goal of CPT is to adapt the backbone model to humor-relevant discourse. We trained Qwen2.5-VL-7B-Instruct with LoRA adapters (rank 32, scaling factor 64, dropout 0.05) and Qwen2.5-VL-32B-Instruct with LoRA layers and 4-bit quantization (rank 8, scaling factor 16) on our curated humor corpus (§A). Training the 7B model ran for 50 epochs with a batch size of 2 and learning rate $5 \times 10^{-5}$ on a single NVIDIA V100 GPU, completing in ∼1 day. Similarly, training the 32B model for 25 epochs with a batch size of 1 and learning rate 1.0e-4 on 4 H100 GPUs required ∼6 hours. Due to our pretraining corpora consisting mostly of textual data, we freeze the entire visual

Table B.1: **Training hyperparameters across stages.**

| Stage | Base Model | Adapter | Epochs | Batch | LR | Prec. | GPUs | Dur. |
|---|---|---|---|---|---|---|---|---|
| CPT | Qwen2.5-VL-7B | LoRA (rank 32) | 50 | 2 | 5e-5 | FP16 | 1× V100 | ∼1 d |
| SFT | CPT model | LoRA (rank 64) | 7 | 16 | 1e-4 | bf16 | 2× H100 | ∼3 h |
| RL | SFT model | Full model | 5 | 16 (roll.) | 1e-6 | bf16 | 4× H100 | ∼1.5 d |

encoder and the multimodal projection layers. Only the language component is updated using LoRA adapters, which prevents catastrophic forgetting of the initial model's visual understanding. Following the continual pretraining, we observed a decline in the 7B model's ability to follow instructions. To address this, we performed instruction-tuning with `LLaMA-Factory` (Zheng et al., 2024) for 1 epoch using 1,000 samples from Alpaca dataset (Taori et al., 2023; Touvron et al., 2023; Wang et al., 2023). For the 32B model, we did not observe any decrease in its instruction-following ability. Figure B.1 and B.2 shows the loss, learning rate (linear decay for the 7B model and cosine learning-rate decay), and gradient norm.

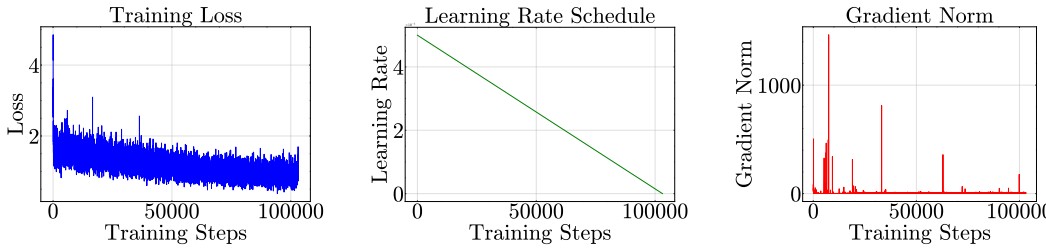

Figure B.1: **CPT dynamics of 7B model.** Loss, learning-rate schedule (linear decay), and gradient norm. Loss decreases steadily; early gradient spikes settle quickly.

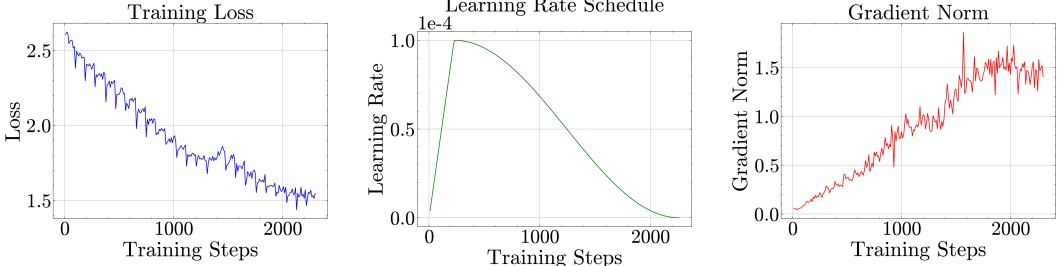

Figure B.2: **CPT dynamics of 32B model.** Loss, cosine LR schedule with warmup, and gradient norm. Loss decreases steadily; later gradient spikes larger.

**Supervised Fine-tuning (SFT).** SFT teaches the model explicit captionist reasoning traces (§3.2). We used the CPT-adapted model as initialization and trained it with `LLaMA-Factory` (Zheng et al., 2024) using LoRA (rank 64 for the 7B model, and rank 8 for the 32B model, all target modules; vision tower frozen). Training utilized our distilled reasoning-trace dataset, running for 7 epochs in bfloat16 precision with an effective batch size of 16. A cosine learning-rate schedule with 10% warmup and peak rate $1 \times 10^{-4}$ was applied. Experiments with the 7B model ran for ∼3 hours on 2×NVIDIA H100 GPUs while for the 32B model, it required ∼6 hours in 4×NVIDIA H100 GPUs. Figure B.3 and Figure B.4 show loss, learning-rate schedule, and gradient norm. Loss drops sharply during warmup and declines smoothly thereafter with stable gradients.

**Reinforcement Learning (RL).** RL aligns model outputs with humor-specific rewards (§3.3). We used EasyR1 (Yaowei Zheng, 2025) with GRPO optimization (DeepSeek-AI et al.,

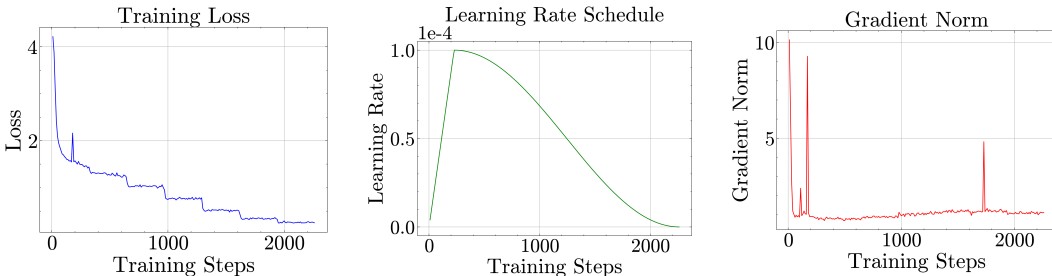

Figure B.3: **SFT dynamics of 7B model.** Loss, cosine LR schedule with warmup, and gradient norm. Warmup produces a sharp loss drop; gradients remain stable.

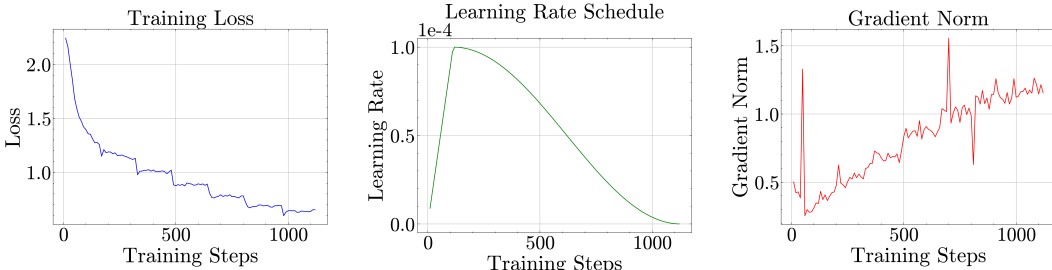

Figure B.4: **SFT dynamics of 32B model.** Loss, cosine LR schedule with warmup, and gradient norm. Warmup produces a sharp loss drop; gradients increase towards to the end of the training.

2025), sampling 3 rollouts per prompt with max length 512 and temperature 1.0. Rollout batch size was 16, learning rate $1 \times 10^{-6}$. Training ran on 4×NVIDIA H100 GPUs (8 for 32B model). Figure B.5 and Figure B.6 report per-batch rewards (EMA-smoothed) for accuracy ($R_a$), format ($R_f$), perception ($R_p$), style ($R_s$), and total reward $R$. As shown, perceptual and style rewards saturate after a few hundred steps, whereas accuracy and format rewards continue to improve steadily. We empirically found that continuing RL training with only accuracy and format rewards beyond this point yields better overall results.

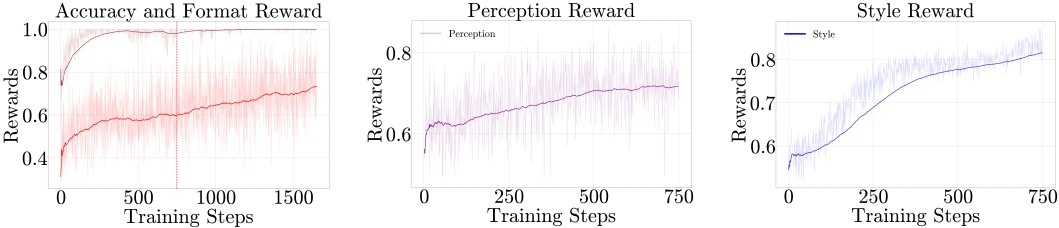

Figure B.5: **RL dynamics of 7B model.** Per-batch rewards for accuracy, format, visual perception, and style reward. The rewards increase as the policy learns humor-aware alignment.

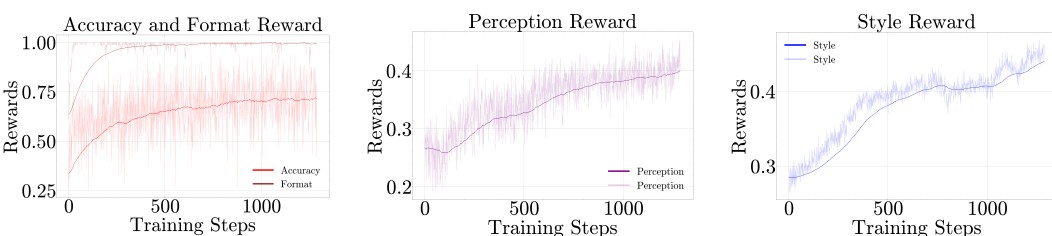

Figure B.6: **RL dynamics of 32B model.** Per-batch rewards for accuracy, format, visual perception, and style reward. The rewards increase as the policy learns humor-aware alignment.

## C  PROMPTS

This section provides the exact prompts used at different stages of our pipeline. To ensure reproducibility, we group them into three: (i) reasoning-trace generation, (ii) reward-judge prompts for RL alignment, (iii) evaluation prompts for text-only and multimodal models.

### C.1  REASONING TRACE GENERATION.

We generate reasoning traces from cartoon annotations of Hessel et al. (2023), which include scene descriptions, uncanny elements, and entity references, providing sufficient grounding.[1] We use DeepSeek-R1 (DeepSeek-AI et al., 2025) as the teacher model, prompted to interpret the visual scene, identify the likely speaker, reconstruct the underlying situation, and analyze humor through incongruity, wordplay, and cultural references before selecting the correct caption among distractors. In rare cases where annotations failed to support the correct caption, the prompt was slightly modified to encourage correct reasoning.

The resulting traces are then rephrased by GPT-4o (OpenAI, 2024) to resemble the style of professional captionist commentary, i.e. concise, observational, and image-grounded. This step replaces description-based phrasing (e.g., *"the description says"*) with image-grounded language (e.g., *"when I look at the cartoon"*), while preserving the underlying reasoning. The aim is to shift the narrative voice from textual paraphrase to multimodal interpretation, ensuring stylistic consistency for downstream training.

Figures C.1 and C.2 shows the generation templates for the matching and ranking tasks, respectively, which structures the reasoning process around scene description, incongruity detection, and caption justification. Figure C.3 shows the rephrasing template, used to refine traces into captionist-style commentary.

---

[1]For the cartoons missing these descriptions, we employ the OpenAI-o3 model OpenAI (2025) to generate them automatically by feeding the cartoon as input.

---

**REASONING TRACE GENERATION PROMPT (MATCHING TASK)**

You are a cartoon analyst evaluating humor in New Yorker Caption Contest. Below is a detailed description of the cartoon, including its visual scene, unusual/uncanny elements, and key observations. Your task is to act as if you're looking directly at the cartoon—not just reading about it. Reason step by step: Think step by step:
1- Understand the visual setting and what makes it strange or surprising.
2- Identify who is most likely speaking in the cartoon.
3- Reconstruct the story or situation behind the scene—what might be going on between the characters?
4- Analyze the humor in each caption: look for metaphors, cultural references, and wordplay.

Finally, from given five captions, one of which matches the cartoon image,
and the others are unrelated, decide which caption that best matches the cartoon image, and justify your choice as if you were analyzing the cartoon visually.

Scene: [SCENE]

Description: [DESCRIPTION]

Uncanny Element: [UNCANNY ELEMENT]

Observations: [OBSERVATIONS]

A) [CAPTION A]
B) [CAPTION B]
C) [CAPTION C]
D) [CAPTION D]
E) [CAPTION E]

Question: Based on what you "see" in the cartoon, which caption best matches the cartoon image? Justify your choice with detailed reasoning based on visual analysis, speaker context, and linguistic play.

{Your answer should support Caption [ANSWER] as the matching caption with strong reasoning.}

Figure C.1: **Prompt for reasoning trace generation (matching).** Transforms cartoon annotations into reasoning traces for caption matching. Correct answer is optionally given.

```
REASONING TRACE GENERATION PROMPT (RANKING TASK)

You are a cartoon analyst evaluating humor in New Yorker Caption Contest. Below is a detailed description of the
cartoon, including its visual scene, unusual/uncanny elements, and key observations. Your task is to act as if you're
looking directly at the cartoon—not just reading about it. Reason step by step: Think step by step:
1- Understand the visual setting and what makes it strange or surprising.
2- Identify who is most likely speaking in the cartoon.
3- Reconstruct the story or situation behind the scene—what might be going on between the characters?
4- Analyze the humor in each caption: look for metaphors, cultural references, and wordplay.

Finally, decide which caption is funnier, and justify your choice as if you were analyzing the cartoon visually.

Scene: [SCENE]

Description: [DESCRIPTION]

Uncanny Element: [UNCANNY ELEMENT]

Observations: [OBSERVATIONS]

A) [CAPTION A]
B) [CAPTION B]

Question: Based on what you "see" in the cartoon, which caption is funnier? Justify your choice with detailed
reasoning based on visual analysis, speaker context, and linguistic play.

{Your answer should support Caption [ANSWER] as the funnier caption with strong reasoning.}
```

Figure C.2: **Prompt for reasoning trace generation (ranking)).** Transforms cartoon annotations into reasoning traces for caption ranking. Correct answer is optionally given.

```
REASONING TRACE REPHRASING PROMPT (MATCHING & RANKING TASKS)

You are an assistant tasked with lightly rewriting a cartoon analysis. Below is a response that was written based on
a textual description of a cartoon. Your ONLY task is to rephrase any language that refers to "the description", "I
imagine", or "visualizing" — and instead replace such phrases with observational ones like "When I look at the
cartoon" or "In the image I see".

- Do not change anything else.
- Keep all reasoning, sentences, structure, and wording identical — only modify phrases that imply the analysis
was based on a description.

--- ORIGINAL RESPONSE ---
[ORIGINAL RESPONSE]

--- REWRITTEN RESPONSE ---
```

Figure C.3: **Prompt for reasoning-trace rephrasing.** Rewrites traces for stylistic consistency.

## C.2 REWARD-JUDGE PROMPTS

We design two LLM-as-judge templates that produce automatic, fine-grained reward signals during RL alignment. Both return binary scores for each criterion, which are aggregated into a reward function (§3.3). At a high level, these rewards operationalize the two key dimensions of humor understanding: (i) perceptual grounding, ensuring that reasoning connects to concrete visual incongruities in the cartoon, and (ii) stylistic fidelity — ensuring that captions and explanations resemble the linguistic qualities of professional humor writing.

**Visual Perception Judge.** This prompt ensures that model reasoning explicitly grounds its explanation in salient visual details. Each cartoon is associated with up to ten curated reference descriptions (entities, background elements, incongruities). Qwen2.5-7B-Instruct receives both the model's reasoning and these references, and outputs a binary vector indicating whether each reference is correctly reflected in the reasoning. Figure C.4 shows the exact prompt. This design ties the perception reward ($R_p$) directly to the visual anchors curated in our dataset (Figure 3).

---

**VISUAL PERCEPTION REWARD PROMPT**

You are an evaluator. Read VISUAL INFO (XML) and the CANDIDATE answer. For each listed tag, output ONLY <tag>0</tag> concatenated in the same order, with no spaces or extra text.
VISUAL INFO: [VISUAL INFO]

CANDIDATE: [ANSWER]

Evaluation rule:
- Output <tag>1</tag> if the candidate explicitly and correctly reflects the fact in VISUAL INFO for that tag.
- Output <tag>0</tag> otherwise (missing, wrong, or contradicted).
Return ONLY these fields concatenated in EXACTLY this order (no spaces/newlines)

---

Figure C.4: **Prompt for visual perception judge.** Evaluates whether reasoning traces explicitly reference curated visual details; outputs binary tags for each attribute.

**Style Judge.** This prompt evaluates the linguistic quality of captions and explanations. Drawing on captionist guidelines (Wood, 2024), the template checks five stylistic dimensions:

1. **Natural phrasing** (use of idiomatic, everyday expressions),

2. **Punctuation** (effective, not missing or overused),

3. **Wordplay** (puns, double meanings, playful twists),

4. **Metaphor** (figurative expressions tied to the cartoon),

5. **Punchline placement** (delivering payoff at the end).

The judge outputs a sequence of binary values corresponding to these criteria, without additional commentary. Figure C.5 shows the full template. The aggregated score forms the style reward ($R_s$), encouraging models not only to choose the correct caption but to justify it in a way that reflects the editorial tone of professional humor writing.

---

**STYLE REWARD PROMPT**

You are an evaluator. Read the CANDIDATE answer and evaluate the caption that is selected as the answer. For each listed tag and each style criterion, output ONLY <tag>0 or 1</tag> concatenated in the same order, with no spaces or extra text.

CANDIDATE: [CANDIDATE]

Evaluation rules:
Determine the caption that is selected for the cartoon. Then, for each of the following style criteria, output <tag>1</tag> if the caption meets the criterion, and <tag>0</tag> otherwise:

<daily_phrase>: 1 if the candidate uses a natural, common, everyday expression or idiom that relates to the cartoon.

<punctuation>: 1 if punctuation is used effectively (not missing, not overused, contributes to readability or style) and enhances the connection to the cartoon.

<wordplay>: 1 if the candidate shows creative wordplay (puns, double meanings, playful twists) that are relevant to the cartoon.

<metaphor>: 1 if the candidate includes a clear metaphorical expression that relates directly to the cartoon.

<punchline>: 1 if the candidate includes a punchline at the end of the caption.

Otherwise output 0.

Return ONLY the tags concatenated in EXACTLY this order with no spaces or newlines:

---

Figure C.5: **Prompt for style judge.** Evaluates captions against stylistic criteria such as phrasing, punctuation, wordplay, metaphor, and punchline placement; outputs binary tags for each dimension.

Collectively, these LLM-as-judge rewards complement the correctness- and format-based signals, enriching the reinforcement learning stage with humor-aware evaluation dimensions.

## C.3  Text-only Evaluation

We evaluate DeepSeek-R1 (DeepSeek-AI et al., 2025) as a text-only baseline using the structured cartoon annotations from Hessel et al. (2023). These annotations provide scene descriptions, uncanny elements, and key observations without requiring access to the original images. For the evaluation, we used separate prompts used in Hessel et al. (2023) for the matching and ranking tasks: the matching prompt presents five candidate captions with one correct option, while the ranking prompt presents two captions with one preferred by the crowd. In both cases, the model is instructed to reason only from the textual annotations. Figures C.6 and C.7 show the full templates.

---

**TEXT-ONLY EVALUATION PROMPT (MATCHING TASK)**

In this task, you will see a description of an uncanny situation of a cartoon from the New Yorker Cartoon Caption Contest. Then, you will see five jokes — only one of which was written about the described situation. Pick which of the five choices truly corresponds to the described scene.

The image takes place in the following location:

Image Description: [IMAGE DESCRIPTION]

Image Uncanny Description: [UNCANNY DESCRIPTION]

The scene includes: [SCENE DESCRIPTION]

One of the following funny captions is most relevant to the scene:
A) [CAPTION A]
B) [CAPTION B]
C) [CAPTION C]
D) [CAPTION D]
E) [CAPTION E]

The funny caption that matches the scene is:

---

Figure C.6: **Prompt for text-only evaluation (matching).** Evaluates DeepSeek-R1 using textual annotations of the cartoon; model selects the caption that best matches the described scene.

---

**TEXT-ONLY EVALUATION PROMPT (RANKING TASK)**

In this task, you will see a description of an uncanny situation of a cartoon from the New Yorker Cartoon Caption Contest. Then, you will see two jokes that were written about the situation. One of the jokes is better than the other one. Pick which of the two jokes is the one rated as funnier by people.

The image takes place in the following location:

Image Description: [IMAGE DESCRIPTION]

Image Uncanny Description: [UNCANNY DESCRIPTION]

The scene includes: [SCENE DESCRIPTION]

A) [CAPTION A]
B) [CAPTION B}

The funnier is:

---

Figure C.7: **Prompt for text-only evaluation (ranking).** Evaluates DeepSeek-R1 using textual annotations; model selects the funnier caption between two options.

## C.4  Multimodal Evaluation

We standardize evaluation of vision–language models using task-specific prompts that present the cartoon image (`<image>`) together with candidate captions. All models are required to respond in the controlled format `<think>...</think><answer>...</answer>`, which enforces explicit reasoning traces while enabling automatic parsing and reward computation. Figures C.8 and C.9 show the full templates.

---

**MULTIMODAL EVALUATION PROMPT (MATCHING TASK)**

[CARTOON IMAGE]

The image is a cartoon from the New Yorker Cartoon Caption Contest. I will provide you with five captions, one of which matches the cartoon image, and the others are unrelated. Choose the caption that best matches the cartoon image:
A) [CAPTION A]
B) [CAPTION B]
C) [CAPTION C]
D) [CAPTION D]
E) [CAPTION E]

First think about the reasoning process in the mind and then provide the answer. The reasoning process and answer are enclosed within <think> </think> and <answer> </answer> tags, respectively, i.e., <think> reasoning process here </think><answer> answer here </answer>

---

Figure C.8: **Prompt for multimodal evaluation (matching).** Cartoon image plus five candidate captions (A–E). Model outputs reasoning and final choice in `<think>` and `<answer>` tags.

---

**MULTIMODAL EVALUATION PROMPT (RANKING TASK)**

[CARTOON IMAGE]

The image is a cartoon from the New Yorker Cartoon Caption Contest. I will provide you with two captions; one of them is deemed funnier by people. Which of the following captions is funnier:
A) [CAPTION A]
B) [CAPTION B}

First think about the reasoning process in the mind and then provide the answer. The reasoning process and answer are enclosed within <think> </think> and <answer> </answer> tags, respectively, i.e., <think> reasoning process here </think><answer> answer here </answer>

---

Figure C.9: **Prompt for multimodal evaluation (ranking).** Cartoon image plus two candidate captions (A–B). Model outputs reasoning and final choice in `<think>` and `<answer>` tags.

To ensure fair comparison, the same prompting protocol is applied to all competing general-purpose multimodal reasoning models (e.g., GLM-4V, Qwen2.5-VL, Kimi-VL). In each case, models are explicitly instructed to *"think before answering"* and to format their outputs in the standardized schema. This guarantees that performance differences reflect reasoning ability rather than prompt design.

# D    ADDITIONAL EXAMPLES: GROUND TRUTH TRACES, EVOLUTION OF MODEL REASONING, AND JUDGE RESPONSES

To complement the quantitative results in the main paper, we provide additional qualitative examples in this section. These examples highlight how ground-truth traces, model outputs, and automated reward signals interact to shape humor-aware reasoning.

## D.1    GROUND-TRUTH TRACES

In addition to the matching example in the main text (Figure 2), we include a ground-truth reasoning trace for a ranking task. As illustrated in Figure D.1, The trace highlights how professional captionists justify one caption over another, often by pointing out which option leverages the visual incongruity more effectively or delivers a sharper punchline. These traces serve as the supervision signals distilled during SFT.

When I look at the cartoon, the setting is an airplane's economy cabin with regular passengers doing typical things—reading, sleeping, wearing headphones. The uncanny element is a giant amoeba-like germ creature sitting in one of the seats, even buckled in. It's massive, with eye-stalks and tendrils, spilling into adjacent seats and the aisle. The man next to it is gesturing toward it, probably talking to the woman by the window. People nearby look uncomfortable, especially the guy behind whose personal space is invaded by the amoeba's feelers.

The key observations: the germ is human-sized, which is absurd since germs are microscopic. It's treated as a regular passenger, which creates humor through juxtaposition. The man in the middle seems to be acknowledging the creature, maybe complaining or making a comment. The humor likely stems from the contrast between the mundane airplane setting and this bizarre, intrusive presence. Next, identifying the speaker. The man in the middle seat is gesturing towards the amoeba while talking to the woman at the window. So he's probably the one delivering the caption. His body language suggests he's making a remark about the creature, possibly annoyed or trying to be humorous. Looking at the captions.

Caption A: "I guess that viral marketing works after all." This plays on the word "viral" which has a double meaning—both marketing that spreads rapidly and relating to viruses/germs. Since the amoeba represents a germ, the pun connects the creature to a viral marketing campaign. It's clever because it ties the absurd visual (a giant germ on a plane) to a common business term.

Caption B: "This guy's wife lets him drink on the plane!", it's implying the amoeba is drunk or its liquid form is due to alcohol. The humor comes from personifying the germ as a rowdy passenger whose wife permits him to drink, leading to his amorphous, spreading state. It's a play on the creature's blob-like appearance resembling someone who's had too much to drink.

Comparing the two. Caption A's strength is the wordplay with "viral" directly linking the germ to a punny scenario. It's witty and fits the visual of a germ being a "viral" entity in a marketing context. However, it might require a bit of cultural knowledge about marketing terms, which isn't universal. Caption B uses physical humor. The amoeba's slimy, spreading form looks like a drunk person spilling over seats. The idea of blaming the wife adds a relatable marital jab. It's more physical and visual, as the creature's appearance directly suggests intoxication.

Which is funnier? Caption B feels more immediate. The visual of the amoeba's gelatinous body spilling over seats perfectly matches a drunk passenger. The marital humor ("wife lets him") adds a layer of relatability. It doesn't rely on wordplay as much as a visceral, silly image. The man gesturing annoyed fits the caption—he's complaining about the creature's behavior, blaming its wife. It's accessible and ties directly to what we see: the creature's form and intrusion. Caption A is clever but more abstract. The "viral marketing" pun is smart, but it's a conceptual leap. The scene shows discomfort and intrusion, not marketing. So Caption B's humor is more integrated with the visuals, making it funnier in this context.

Figure D.1: **Example reasoning trace for a caption ranking task.** Given a cartoon and two candidate captions, the model is asked to identify which one is judged funnier. The reasoning trace explains the visual setup (a giant amoeba crowding an airplane seat), evaluates both captions with respect to the scene, and justifies why Caption B ("This guy's wife lets him drink on the plane!") better matches the visual humor. The explanation highlights how ranking traces capture comparative judgments of wit versus visual grounding, training the model to reason in captionist-style trade-offs.

## D.2 Evolution of Model Reasoning.

To examine how our three-stage pipeline shapes reasoning, we compare outputs from the SFT-only and SFT+RL models on both matching and ranking tasks.

Figure D.2 presents a caption matching example. The SFT-only model generates fluent but superficial descriptions, often framing the parrots' setup as a "therapy session" without fully tying this to the incongruity at the core of the humor. Its explanations are verbose, repetitive, and miss layered cues. In contrast, the SFT+RL model grounds its reasoning in salient visual evidence, filters irrelevant distractors, and highlights multi-layered mechanisms—species-appropriate wordplay, speaker positioning, and ironic juxtapositions. The output reads closer to a captionist's commentary: concise, well-structured, and attuned to comedic effect.

Figure D.3 highlights a ranking task, where the contrast becomes even sharper. The SFT-only model tends to fixate on literal associations (e.g., coffee grounds as a clue of recency) while overlooking the broader absurdity of cowboys encountering modern espresso machines. The SFT+RL model instead integrates the scene's anachronism with stylistic cues, judging which caption better captures the visual gag without relying on niche jargon. Here again, reinforcement learning with perceptual and stylistic rewards enables the model to move beyond surface plausibility toward humor-aware reasoning.

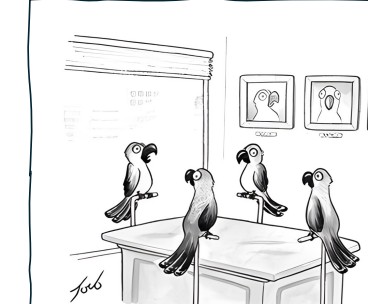

QUESTION

Which of the five choices is the matching caption?

CAPTION CHOICES

A) Unfortunately, your time is up, but I think we have a general outline of your problem.
B) There are cheaper ways to find an audience, Jim.
C) Can we talk?
D) It's our new interactive TV. When he's done, I'm having an appendectomy on 'General Hospital.'
E) I hope Skip doesn't ask me to sacrifice.

SFT-ONLY MODEL RESPONSE

The cartoon shows four parrots in a doctor's office. Each has a microphone or perch, and they're looking at framed portraits on the wall. The uncanny element is that the parrots appear to be mimicking human behavior—taking turns looking at their own portraits, as if they're patients in a therapy session. When I look at the cartoon, the setting is a doctor's office with typical elements like an examination table and blinds. The portraits are of the parrots themselves, which is unusual. The humor comes from the absurdity of parrots acting like humans, especially in a therapy context. In the image, one parrot is likely speaking, possibly commenting on the situation. The others are engaged in looking at the portraits. The story seems to be a therapy session where the parrots are analyzing their own images, which is funny because they're birds, not humans.

Now, analyzing each caption:
A) Unfortunately, your time is up, but I think we have a general outline of your problem. This sounds like a therapist ending a session. The parrots are in a therapy setup, so this fits. The humor is in applying human therapy to parrots.
B) There are cheaper ways to find an audience, Jim. This might refer to publicizing something, but it doesn't connect to the therapy or portraits. The parrots aren't showing audience interest, so it's unrelated.
C) Can we talk? A common phrase, but here it could be a parrot speaking. However, it's too vague. The therapy context is implied, but the caption is too generic without tying into the portraits.
D) It's our new interactive TV. When he's done, I'm having an appendectomy on 'General Hospital.' This mentions TV and surgery, which doesn't relate to the therapy portraits. The parrots aren't watching TV; they're looking at their own portraits. So, it's off-topic.
E) I hope Skip doesn't ask me to sacrifice. This might imply a religious or ritual context, like sacrificing something. The parrots are birds, and "sacrifice" could be a pun, but it doesn't fit the therapy scene. The portraits aren't related to sacrifice.

Comparing all, caption A fits best. The therapist ending a session with parrots makes sense, and the humor comes from applying human therapy to animals. The portraits are part of the therapy, so the caption ties everything together. The other captions don't connect as well, either because they're off-topic or too vague.

SFT+RL MODEL RESPONSE

In the cartoon, I see a corporate office setting with four parrots sitting on stools around a conference table - three facing forward and one turned sideways like a facilitator. Above them are framed self-portraits of another parrot pair. The absurdity hits immediately: birds replacing humans in what should be a human professional environment, with those portraits suggesting this isn't just any office but one where birds literally document their colleagues' "faces" (pun intended).

Looking at the captions:
- Option A feels too human-centric (therapist/human patient scenario)
- Option B implies an audience/celebrity context (no visual cues like microphones/stages)
- Option C suggests personal conversation (only group observation shown)
- Option D introduces medical/TV chaos (absent visual elements)
- Option E hints at sacrificial themes (no predators or threats visible)

Option C stands out perfectly - that sideways parrot could easily be "facilitating" a meeting while others listen! The phrase "Can we talk?" visually mirrors their clustered posture and the meeting context. The humor lands through triple-layered irony: (1) Birds as corporate workers (absurd premise), (2) Parrot language mimicking human small talk (species-appropriate gag), (3) The speaker's position (facilitator/observer) determines who says it. Every visual detail serves the joke: portraits establish "office culture," perches show hierarchy, and the unspoken tension makes "Can we talk?" feel live. The winning caption weaponizes every visual gag point while letting the image breathe. 🎯👍

Figure D.2: **Example reasoning on a caption matching task.** The SFT-only model provides fluent but superficial descriptions, framing the parrots' scene as a "therapy session" while missing deeper incongruities. The SFT+RL model, by contrast, grounds its reasoning in salient visual cues and highlights layered humor mechanisms (species-appropriate wordplay, speaker positioning, and irony), producing commentary closer to professional captionist style.

These examples mirror our quantitative findings: RL alignment not only improves accuracy but also transforms the style of reasoning, shifting outputs from literal, mechanical

○

**QUESTION**

Which caption is funnier than the other one?

**CAPTION CHOICES**

A) Grounds still warm...
B) Follow the plastic.

**SFT-ONLY MODEL RESPONSE**

The scene is a desert with two cowboys on horseback. They've just come across a crashed coffee machine and scattered coffee items. The uncanny element is the juxtaposition of a modern espresso machine in the Old West.

The cowboys are likely commenting on the absurdity of finding such a device in the desert. The captions need to tie into the visual surprise.

Caption A: "Grounds still warm..." plays on the coffee grounds being nearby, implying the machine crashed recently. It's a literal take but might be too straightforward.
Caption B: "Follow the plastic" references the machine's plastic parts among cacti. It's more absurd, suggesting they're following a scent trail made of plastic. This adds a layer of silliness with the cowboys treating the machine debris like a treasure hunt.

The humor in B is stronger because it extends the absurdity visually. The answer: B.

**SFT+RL MODEL RESPONSE**

In the cartoon, there are two cowboys in a desert, surrounded by a broken espresso machine, coffee grounds, and other coffee-making tools. They're far from any civilization, yet they're set up like they're in a campsite. The absurd element jumps out immediately—espresso in the Wild West? That anachronistic contrast is the core joke.

The cowboys' bewildered expressions heighten the absurd contrast. The humor lies in this surreal mix of frontier life and modern tech.

Caption A works perfectly because "Grounds still warm" plays on coffee metaphors (fresh beans) while mirroring desert heat (metaphors "hot" coffee vs "hot" environment). No need for niche jargon—just pure visual metaphor clicking while matching the art absurdism without needing niche jargon.
Caption B's "plastic" metaphor is clever tech wordplay but demands niche knowledge (ESPRESSO pods?) while lacking in desert/desert wordplay. Without such tech jargon, B falls flat without same surreal absurdism without jargon overkill.

Winner: A nails it visually and simply while being instantly readable like a cowboy one-liner while looking exactly like cartoon chaos without needing niche jargon. Perfect match! Answer A.

Figure D.3: **Example reasoning on a caption ranking task.** The SFT-only model fixates on literal associations (e.g., coffee grounds as evidence of recency), while the SFT+RL model integrates the broader absurdity of modern technology in a cowboy setting. Reinforcement learning with perceptual and stylistic rewards enables humor-aware judgment that avoids niche jargon and reflects the cartoon's visual gag.

justifications toward the layered, economical, and culturally grounded tone of professional captionists.

### D.3    JUDGE RESPONSES

Figures D.4 and D.5 illustrate the outputs of the LLM-as-judge components used to compute the perception and style rewards, respectively. In Figure D.4, the perception judge compares the model's explanation against curated visual references and returns per-attribute binary scores, along with an aggregate score; this verifies that reasoning is explicitly grounded in salient scene details. In Figure D.5, the style judge resolves the model's <answer> to the selected caption and evaluates it along five binary dimensions, namely daily phrasing, punctuation, wordplay, metaphor, and punchline, providing a structured signal for humor-aware linguistic quality.

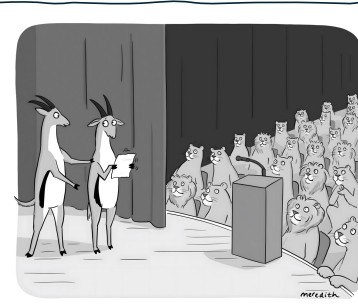

VISUAL REFERENCES

Info 1. Two upright, bipedal goats standing on a conference stage
Info 2. One goat holding and reading from a visibly trembling sheet of paper.
Info 3. The second goat positioned beside the first, oriented toward it in a posture.
Info 4. A podium placed on the stage beside the two goats.
Info 5. An audience composed entirely of seated lions facing the stage.
Info 6. All lions sharing identical wide-eyed, vacant facial expressions.
Info 7. Lions sitting upright with paws folded in a human-hand-like manner.
Info 8. A microphone positioned on the stage between the goats and the unused.
Info 9. A background curtain and curved table indicating a formal auditorium setting.
Info10. Lions arranged in symmetrical seating rows with uniform posture and expression.

MODEL RESPONSE

When I look at the cartoon, the scene shows two goats on a stage addressing an audience of lions in theater seats. What makes this absurd is that prey animals are giving a speech to predators, which is essentially the opposite of natural behavior. The visual contrast is heightened by the fact that the lions are sitting calmly while the goats appear nervous, holding notes.

Now, examining the captions.

Caption A ("Just pretend none of them are wearing their teeth") creates a dark humor interpretation where the teeth imply the lions might eat the goats. This requires the viewer to imagine, which adds a layer of unease. However, I notice that in the image, the lions look cheerfully attentive rather than menacing, which makes this interpretation feel a bit forced.

Caption B ("The good news is that you have their full attention") works better because it directly plays off the visual irony. The phrase "their full attention" suggests the speaker (likely a goat) is commenting on how captive the lions appear, while actually implying the lions are **paying too much attention** to them. This creates a witty double meaning - professionally "full attention" means concentration, but but in this context it hints at the predator's curiosity.

The visual cues support Caption B more effectively. The lions' uniform, mesmerized expressions resemble an audience at a motivational talk rather than their natural predatory behavior. Caption B's dry delivery matches the goats' anxious body language while highlighting the absurdity through understatement. It transforms the scene from potentially scary to unintentionally funny.

Caption A's imaginative twist feels disconnected from what we actually see - the lions' expressions suggest admiration, not aggression. The humor works best when amplifying existing visual contrasts without inventing new threats. Caption B achieves this by essence by focusing on the core joke: prey giving speeches to predators.

Answer: B

JUDGE RESPONSE (VISUAL PERCEPTION REWARD)

Info 1: 1.0, Info2: 1.0, Info 3: 1.0, Info 4: 1.0, Info 5: 1.0, Info 6: 1.0, Info 7: 0.0, Info 8: 1.0, Info 9: 1.0, Info 10: 1.0
Total Score: 9.0

Figure D.4: **Visual perception judge response.** Binary scoring of whether model reasoning reflects curated reference details.

## D.4 COMPARISON WITH EXPERT TRACES

For comparison with an expert, we selected an example where the model's final choice differs from that of the human expert. As shown in Figure D.6, the model nonetheless reproduces the same core interpretive steps–identifying the salient incongruity, evaluating competing narrative readings, and articulating a justification. This alignment in *reasoning structure*, despite a different conclusion, suggests that the model has internalized the captionist analytic framework rather than merely learning surface-level patterns.

## D.5 DISCUSSION

The additional examples presented in Appendix D provide more than anecdotal evidence. They open a window into how our pipeline achieves humor alignment. Ground-truth traces illustrate the kinds of structured signals distilled into the model during supervised fine-tuning, making explicit the captionist-style scaffolds that guide learning. The staged outputs across CPT, SFT, and RL reveal how reasoning evolves: from surface-level associations to visually grounded, stylistically fluent explanations. Finally, the judge responses show

---

EVALUATION RULES

- Daily Phrase: 1 if the caption uses a natural, common, everyday expression or idiom.
- Punctuation: 1 if punctuation is used effectively (not missing, not overused, enhances readability or style).
- Wordplay: 1 if there is creative wordplay (puns, double meanings, playful twists) relevant to the cartoon.
- Metaphor: 1 if a clear metaphorical expression relates to the cartoon.
- Punchline: 1 if the caption includes a punchline at the end.
- Otherwise, 0.

SAMPLE MODEL RESPONSE FAVORING A STYLISTICALLY NICE CAPTION

The cartoon shows two people reading a book together.

…

Of the options, A) 'I guess we're not on the same page… literally.' is the best match because it plays on the idiom and the literal situation.

Answer: A

JUDGE RESPONSE (STYLE REWARD)

Daily Phrase: 1, Punctuation:1, Wordplay: 1, Metaphor: 0, Punchline:1

SAMPLE MODEL RESPONSE FAVORING A STYLISTICALLY BAD CAPTION

The cartoon shows a man holding a book.

…

Of the options, B) 'This book is very interesting.' best matches the cartoon.</think>

Answer: B

JUDGE RESPONSE (STYLE REWARD)

Daily Phrase: 0, Punctuation: 1, Wordplay: 0, Metaphor: 0, Punchline: 0

---

Figure D.5: **Style judge response.** Binary scoring of captions across natural phrasing, punctuation, wordplay, metaphor, and punchline placement.

the mechanics of reinforcement learning in action, translating abstract notions of "wit" or "grounding" into quantifiable feedback signals.

Taken together, these materials highlight the transparency of our approach. They demonstrate that reported gains are not only measurable in accuracy, but also interpretable in terms of reasoning quality, alignment with visual anchors, and stylistic fidelity. By surfacing these intermediate signals, we highlight a broader contribution: humor is not treated as a black box but as a process that can be decomposed, supervised, and aligned. This transparency is essential if computational humor is to serve as a reliable probe of multimodal reasoning.

## E  LIMITATIONS AND ETHICAL CONSIDERATIONS

While our framework advances humor understanding in multimodal LLMs, several limitations remain.

**Visual perception errors.** Even after RL alignment, the model can misread salient actors or props, which cascades into faulty reasoning and wrong caption selection. Figure E.1 illustrates a typical failure: the foreground figure is misidentified as the Grim Reaper (scythe), whereas the humor relies on the incongruity of a Viking warrior in a domestic kitchen. Our 32B CPT+SFT+RL model correctly identifies the visual setup, suggesting that scaling the backbone meaningfully improves perceptual grounding and may reduce such cascading reasoning failures. A key remaining bottleneck, however, is the vision encoder itself: stylized cartoons require abstractions that are underrepresented in current pretraining corpora. While our perception reward partially compensates for this mismatch, substantial progress will likely require larger, publicly available cartoon datasets that better capture the visual conventions of editorial humor.

**Shallow cultural grounding.** The model sometimes produces fluent but brittle analyses that miss culturally embedded references. As shown in Figure E.2, an SFT-only

**QUESTION**

**Which caption is funnier?**
A) They say cliche- I say classic.
B) You're gonna KILL in that outfit.

**MODEL RESPONSE**

In the cartoon, I see why this is funny. Looking at the image: a guy in what looks like formal wear—suits, ties, typical shopping vibe. But instead of humans modeling clothes, we have these... creatures. Like, horned or monster-like. That visual irony—expecting humans but seeing these weird animal hybrids. Their postures mimic how models pose seriously, but their appearances are anything but classic. Those sharp horns and unnatural stances contrast hilariously with the idea of fashion critique. Caption B ("You're gonna KILL in that outfit") works on three levels:
1) Visual irony: The "monster" models literally look ready to kill 2) Fashion metaphor: High praise for clothing 3) Double entendre: "Kill" as both compliment and literal threat from their appearance
Caption A misses these layers:
- "Cliche vs classic" doesn't connect with visual monsters
- No wordplay with their physicality - Misses the visual metaphor Caption B maximizes humor through:
Visual metaphor: Monster-like figures "modeling"
Double entendre: Fashion praise vs literal killing
Wordplay: "Kill" as both compliment and visual threat
Irony: Calling monstrous figures "stylish" Caption B uniquely maps to:
1) Monster appearance ("kill")
2) Fashion context ("outfit")
3) Double meaning of "kill"
4) Visual parody of high-end retail
This preserves caption contest tradition of naming banal visual irony. Caption B weaponizes every visual element: unnatural poses + monster aesthetics + fashion metaphor + literal wordplay. Caption B weaponizes visual irony precisely while caption A wastes visual cues." This preserves caption contest tradition of naming banal visual irony. Caption B maximizes visual-fantasy-combination.

**HUMAN EXPERT RESPONSE**

We are in what looks like a high-end costume or clothing shop for predators. A wolf stands in front of a mirror wearing a full sheepskin suit, complete with fluffy wool like a tailored garment. A salesman stands nearby holding a zebra skin as another possible outfit. Around them are other garments:
- A cowhide draped over a chair
- A goat outfit
- An antelope

Other animal skins, all as though they were merchandise The wolf is clearly pleased with his choice, the salesman seems to have critiqued it, as if he's a stylist trying to steer the client towards something more original. The caption: "You say cliche, I say classic." In everyday life, a salesperson at a boutique might say, "You say cliché, I say classic," when defending a timeless but overused look. Here, he's not defending it; he's characterizing it as a boring, obvious look with no originality. For the sheep, disguise is not about originality. It's about functionality. Only as a "wolf in sheep's clothing" can he infiltrate a flock and successfully prey on them.

A zebra or a cow disguise might be more creative visually, but it would hardly be useless for the wolf's actual goal. Just imagining a wolf in a cow's clothing or a zebra's clothing is funny in and of itself. It's part of the scene's background humor.

Ultimately, the humor comes from this mid-match. The salesman is judging the choice by aesthetic originality, while the wolf and the viewer know that for hunting sheep, this cliché outfit fit the situation to a T.

Figure D.6: **An example where the model and the expert reach different conclusions while following a similar thought process.**

model latches onto surface wordplay (*"plane"*) and ignores the cartoon's Superman motif—a popular-culture cue. After RL, the model gestures toward the pop-culture context but over-attributes visual evidence (e.g., inventing a *"flying-pose"* link) and still fails to articulate the catchphrase-based joke. This illustrates the challenge of grounding cultural knowledge without hallucination.

**Model size and scaling.** Our experiments use a 7B-parameter backbone (Qwen2.5-VL-7B). While this provides a fair and reproducible open baseline, it naturally underperforms large proprietary systems such as o3 and GPT-4o. Our focus in this work is not raw scale, but the development of humor-specific priors, reasoning traces, and reward functions. We also scaled our pipeline to a larger open backbone, Qwen2.5-VL-32B model, and observe the improvement in the performances on matching and ranking tasks as presented in the Table 1.

POST-SFT GENERATED DESCRIPTION (7B MODEL)

The cartoon shows a modern kitchen scene with a woman in a bathrobe holding a breakfast tray. Across from her, there's a man dressed as the Grim Reaper—complete with the scythe, horned helmet, and straw-like hair—but he's just sitting at the table eating a donut like it's totally normal. The absurdity hits immediately because these two couldn't be more different—one's a mundane homebody, the other a mythological horror figure, both acting like it's just another Tuesday.

POST-SFT+RL GENERATED DESCRIPTION (7B MODEL)

In the cartoon, I see a modern kitchen setting with a woman preparing breakfast (wearing bathrobe, holding tray) contrasted by a terrifying Grim Reaper-like figure seated at the table. The core absurdity centers on this supernatural entity casually eating breakfast as if it were normal. Key visual cues include the Reaper's scythe/soup spoon, ragged robes, and menacing horns/hat - all jarring against the mundane domestic backdrop. The woman's paused action (likely mid-carry of breakfast) suggests she's reacting to this surreal intrusion.

POST-SFT GENERATED DESCRIPTION (32B MODEL)

The cartoon depicts a kitchen scene where a woman in a bathrobe is serving breakfast to the Grim Reaper, who's sitting at the table with his scythe. The humor comes from the juxtaposition of a mundane domestic setting with the supernatural figure of Death. First, the visual elements: The Grim Reaper's presence in a kitchen is inherently absurd—he's usually associated with deathbeds, not breakfast tables. The woman's nonchalant attitude amplifies the humor, as if serving Death is a normal occurrence. The scythe on the floor and the checkered tablecloth add to the domestic surrealism.

POST-SFT+RL GENERATED DESCRIPTION (32B MODEL)

In the cartoon, I see why this is funny. Looking at the image: a typical kitchen setup—checkered tablecloth, someone in a robe, maybe having breakfast. But the big guy in the corner... those horns, the scythe, the long hair... classic devil vibes. But here he is, just sitting there, acting normal. That visual irony— expecting something intimidating, but he's... mundane? That's the twist: a scary figure in everyday life. His size contrasts with the ordinary setup—big, horned guy in a tiny kitchen corner while the other seems nonchalant.

**Winning Caption:** *"We're all out of the blood of your enemies. You'll have to settle for marmalade."*

Contest #132

Figure E.1: **Limitation: Visual perception errors.** All 7B-scale variants of our model misidentify the foreground figure as the Grim Reaper (with scythe), while the humor relies on the incongruity of a Viking warrior in a domestic kitchen. This perceptual error cascades into mismatched reasoning and incorrect caption selection. When scaling our approach to a larger model, we observe that it initially misidentifies objects in the early stages of training, however, with our introduced approach, which incorporates two additional rewards in RL setup, the model corrects these shortcomings and begins generating responses that correctly identify the objects.

**Cultural and linguistic specificity.** Humor is highly contextual and culturally grounded. Our focus on the New Yorker Cartoon Caption Contest, predominantly U.S. and Anglophone, limits generalization across cultures, languages, and humor traditions. Extending to multilingual and multicultural humor datasets is an important direction.

**Subjectivity in evaluation.** Even within NYCC, crowd votes and editorial selections capture only partial perspectives on what is funny. Our rewards approximate these preferences through perceptual and stylistic judges, but cannot fully resolve the inherent variability in humor appreciation. Broader human evaluation remains necessary.

**Data access and licensing.** Parts of our continual pretraining corpus are derived from copyrighted sources (e.g., podcast transcripts, published books). Due to licensing restrictions, we cannot release this material directly, though we will provide evaluation splits

**POST−SFT GENERATED DESCRIPTION**

Sounds like a bird....no, a plane!' This plays on Superman's hearing, but it's less connected to the scene. The stethoscope is meant for internal sounds, but the doctor's comment about external noises like planes feels disconnected. It doesn't tie back to the main joke of the pointless exam.

**POST−SFT+RL GENERATED DESCRIPTION**

Caption B works perfectly! The doctor's stethoscope position mirrors Superman's iconic flying pose. That visual pun makes it feel organic. The caption extends the joke - we imagine Superman's powers manifesting as heart sounds. Plus, the doctor's bewildered expression matches the plane revelation. Every visual element supports this. Caption B transforms the image's tension: we wonder about unseen sounds, imagine Superman's reaction. The medical context even justifies the doctor's delivery.

Figure E.2: **Limitation: Shallow cultural understanding.** The humor depends on a Superman reference (e.g., the well-known *"it's a bird... it's a plane..."* trope). The SFT-only model fixates on superficial wordplay; the SFT+RL model recognizes the pop-culture context but hallucinates a visual link and does not fully recover the cultural script.

and full preprocessing pipelines. This ensures reproducibility while respecting intellectual property.

Despite these limitations, our work represents a step toward aligning multimodal reasoning models with one of the most elusive facets of human intelligence: humor. Addressing cultural, subjective, and computational challenges offers fertile ground for future research.

# F  CROSS-DATASET GENERALIZATION

Although our model is trained exclusively on New Yorker Cartoon Caption Contest (NYCC) data, we additionally assess its ability to generalize across humor domains that differ substantially in visual structure, linguistic form, and reasoning requirements. We perform zero-shot experiments on two external datasets:

- **YesBut** Hu et al. (2024): A two-panel visual humor dataset centered on contrastive reasoning ("yes...but..."), which differs from NYCC's single-image incongruity-resolution format. Among its four subtasks, only the two classification-based ones ("Philosophy" and "Title") are directly comparable to caption-selection setups, so our evaluation focuses on these tasks.

- **DeepEval** Yang et al. (2024): A broad multimodal evaluation benchmark. Only 2.9% of its images belong to a humor subset. Tasks measure semantic alignment, descriptive accuracy, and title appropriateness—none are cartoon-captioning–specific.

Despite being trained only on NYCC, our model shows strong cross-domain transfer. On YesBut, it achieves large absolute gains over its base model (+31 to +34 points, Table F.1). On DeepEval, it again exhibit strong improvements on semantic and descriptive judgments (Table F.2), even though the dataset contains minimal stylistic overlap with NYCC.

Table F.1: **Zero-shot Generalization to YesBut (Hu et al., NeurIPS 2024).** Accuracy (%) on the Philosophy and Title subtasks. Our models are trained only on NYCC data yet show strong transfer, particularly for the 7B backbone.

| Model | Philosophy | Title |
|---|---|---|
| Qwen2.5-VL-7B-Instruct (Base) | 43.19 | 29.11 |
| Ours-7B | **74.90** | **63.32** |

Table F.2: **Zero-shot Generalization to DeepEval (Yang et al., ACL 2024).** Performance on the *humorous* subset (29 images, 2.9% of the benchmark). We report accuracy on three tasks: DeepSemantics, Description, and Title.

| Model | DeepSemantics | Description | Title |
|---|---|---|---|
| Qwen2.5-VL-7B-Instruct (Base) | 10.34 | 24.13 | 34.48 |
| Ours-7B | **54.43** | **100.00** | **63.18** |

These results provide evidence that the proposed reasoning-trace supervision and humor-aware alignment enable the model to acquire transferable multimodal reasoning patterns rather than overfitting to NYCC-specific stylistic templates.

## G  DETAILED ABLATIONS ON MODEL TRAINING STAGES

To further clarify the contribution of each training component in our CPT → SFT → RL pipeline, we report a comprehensive ablation over all combinations of components. Results are given in Table G.1. All experiments use the same 7B backbone and training budgets.

Table G.1: **Ablation of training stages.** Incrementally adding continual pretraining (CPT), supervised fine-tuning (SFT), and reinforcement learning (RL) reveals that SFT provides the main reasoning gains, CPT enhances SFT, and RL further improves correctness and grounding. The full pipeline (CPT + SFT + RL) delivers the best performance across all evaluation settings.

| Approach | (Hessel et al., 2023) | | (Zhou et al., 2025) | |
|---|---|---|---|---|
| | Matching | Ranking | 10-vs-1000 | 30-vs-300 |
| Base | 42.67 | 55.06 | 50.57 | 47.99 |
| Base + CPT | 41.00 | 51.69 | 50.29 | 51.43 |
| Base + SFT | 47.00 | 56.88 | 49.71 | 50.86 |
| Base + CPT + SFT | 49.00 | 56.88 | 54.57 | 49.14 |
| Base + RL | 56.67 | 58.96 | 54.86 | 49.14 |
| Base + SFT + RL | 57.33 | 58.18 | 56.29 | 44.86 |
| Base + CPT + RL | 46.00 | 51.95 | 49.43 | 50.29 |
| Base + CPT + SFT + RL (Ours) | 60.33 | 62.08 | 53.14 | 49.43 |

Our key findings are as follows:

1. **SFT is the primary source of structured reasoning**. Base+SFT yields substantial improvements across all tasks. This confirms that captionist-style reasoning traces are the main driver of the model's ability to analyze incongruity, evaluate alternate interpretations, and justify a final choice.

2. **CPT specifically enhances SFT, not RL**. CPT alone does not improve accuracy, and Base+RL outperforms CPT+RL. CPT provides conceptual prerequisites (editorial heuristics, incongruity resolution patterns, narrative conventions) that are directly leveraged during SFT. RL, by contrast, optimizes correctness, grounding, and format through reward signals and therefore benefits less from such background knowledge. As a result, CPT helps when paired with SFT but can introduce stylistic variance that slightly destabilizes RL when used without reasoning supervision.

3. **The full pipeline is complementary**. The best results arise only when CPT, SFT, and RL are applied together. CPT supplies the conceptual background assumed by captionist reasoning; SFT teaches the structured analytic procedure; RL sharpens correctness and grounding. This division of labor explains the observed ablation trends and the necessity of all three stages for strong humor-reasoning performance.

## H  LLM Usage

LLMs were used to polish the writing and improve the clarity and flow of the text. All final revisions were reviewed and edited by the authors.

