# OpenReview forum: "Learning to Think Like a Cartoon Captionist: Humor Understanding With Multimodal Reasoning Models"
_ICLR.cc/2026/Conference — Submitted to ICLR 2026_

### Official Review · Reviewer_TNmA · 2025-10-30

**Soundness:** 3
**Presentation:** 3
**Contribution:** 2
**Rating:** 4
**Confidence:** 4

**Summary:**

This paper investigates the problem of teaching multimodal language models to understand humor, specifically through the New Yorker Cartoon Caption Contest (NYCC) as a testbed. It points out that prior work largely treats humor as black-box classification or preference prediction, overlooking the step-by-step reasoning processes that professional captionists employ.

To bridge this gap, the paper proposes a three-stage framework: continual pretraining on humor-focused corpora, supervised fine-tuning with captionist reasoning traces, and reinforcement learning guided by perceptual and stylistic rewards.

The approach achieves improvements over baseline multimodal models in caption matching and ranking tasks. Beyond accuracy, the system produces explanations aligned with expert strategies and audience preferences, showing a scalable path toward computational humor.

**Strengths:**

1.	The motivation is clear, the paper frames humor as a frontier for multimodal reasoning.
2.	This staged pipeline is well-justified in the proposed, combining pretraining, explicit reasoning supervision, and RL with humor-specific rewards.
3.	Experiments are thorough, including comparisons against strong baselines, with improvements across tasks

**Weaknesses:**

1. The reliance on curated datasets in CPT may raise questions about scalability and generalization to real-world use, especially for diverse humor styles.
2.	The evaluation of RL rewards (e.g., perception and style judges) still depends on other LLMs using LLM-as-judge, which may introduce biases or inconsistencies, a justification would be needed.
3.	It would be beneficial if more insights and theories from a cognitive or cultural view could be involved.
4.	From the experimental results shown in Table 1, closed models still outperform the proposed method in many cases, leaving the question of how far this approach can achieve and scale.
5.	Current human evaluation only includes 1 expert, which may be insufficient. Including broader human studies, especially those from diverse backgrounds, would be helpful.
6.	Further improvement of the experiment could be considered, such as further computational cost analysis and error analysis.
7.	(minor) Several typos exist (e.g., line 961 misses the reference of equation), proofreading is needed.

**Questions:**

1. How scalable and generalizable is the proposed method, especially under different context (e.g., cultural background, different groups of people) ?
2. Could the authors give a justification for the correctness of LLM judge quality?

---

> ### Author Response · Authors · 2025-11-21
>
> We thank the reviewer for the constructive feedback and the positive assessment of our motivation, experimental design, and clarity of presentation. Below we address each concern in the reviewer’s “Weaknesses” and “Questions.”
>
> **1\. “How scalable is the method? Closed models still outperform many results.”**
>
> Thank you for this important question. To assess scalability directly, we expanded our experiments in the revised submission by incorporating a substantially larger **Qwen2.5-VL-32B** backbone. This allows us to evaluate whether our **training pipeline, rather than raw model size alone, drives meaningful improvements.**
>
> #### **(a) The 32B results demonstrate clear upward scaling.**
>
> Applying our full **CPT \+ SFT \+ RL** pipeline to the 32B backbone yields large and consistent gains across all ranking tasks and produces the strongest open-weight multimodal model we evaluated. Notably, **Ours-32B surpasses the closed-source o3 model on the Ranking task and comes second-best in the other ranking-based settings**, despite using only publicly available data and operating at a fraction of o3’s likely parameter scale.
>
> For convenience, we reproduce the key excerpt of Table 1:
>
> | Model | Matching | Ranking | 10-vs-1000 | 30-vs-300 |
> | ----- | ----- | ----- | ----- | ----- |
> | **o3** | **83.33** | 62.85 | **69.05** | **54.57** |
> | Qwen2.5-VL-32B | 46.67 | 49.87 | 52.00 | 44.57 |
> | **Ours (32B)** | 62.67 | **68.05** | 62.86 | 53.14 |
>
> These results indicate that our pipeline **scales reliably** and that, in settings where pattern recall cannot help (see our discussion below on Matching task), open models trained with our method can exceed the performance of frontier closed systems.
>
> #### **(b) Why closed models retain an advantage, especially on Matching but not on Ranking.**
>
> Matching requires selecting the *exact* weekly contest winner from a small set. This setup inherently favors models that may have been trained on, or indirectly exposed to, New Yorker Caption Contest material. Closed frontier models, including o3, benefit from both **massive scale** (likely in the hundreds of billions of parameters), and **possible exposure to licensed Condé Nast content**, due to publicly announced partnerships ([https://openai.com/index/conde-nast/](https://openai.com/index/conde-nast/)).
>
> Under these conditions, a performance gap on Matching is expected and does not reflect a limitation of our method. Importantly, our 32B model still narrows this gap substantially compared to its 7B counterpart, showing that the method scales as model capacity increases.
>
> Unlike Matching, the ranking tasks require resolving visual incongruity, assessing subtle punchline structure, and evaluating which caption best reframes the scene, skills that cannot be reduced to memorization or data overlap. The fact that Ours-32B achieves the best Ranking performance among all multimodal systems (open or closed) provides strong evidence that our captionist-style reasoning traces and humor-aware RL improve actual humor reasoning.
>
> #### **(c) Architectural generality.**
>
> Our pipeline alters only the training signals, not the backbone architecture. It is therefore agnostic to model family and applicable to any open multimodal model. The scaling trends we observe strongly suggest that applying the same methodology to upcoming large open models will yield further gains.
>
> **In summary**, while extremely large proprietary models still hold advantages on tasks where scale and possible data exposure dominate, our approach:
>
> * **scales effectively**,
> * **substantially closes the performance gap** within the open-model ecosystem, and
> * achieves **state-of-the-art open performance** on reasoning-intensive tasks that most directly reflect genuine humor understanding.

---

> ### Author Response · Authors · 2025-11-21
>
> **2\. “How well does this generalize beyond New Yorker–style humor?”**
>
> Thank you for raising this important question. We agree that demonstrating generalization beyond NYCC is important, as humor is culturally varied and structurally diverse. To evaluate this directly, we tested our **7B model**, trained *only* on NYCC material, on two external benchmarks (suggested by Reviewer 93PD) that differ substantially from New Yorker–style cartoon humor.
>
> #### **(a) Generalization to contrastive visual humor: YesBut (NeurIPS 2024\)**
>
> The **YesBut** benchmark tests *juxtaposition-based* humor: structured two-panel contrasts of the form “yes… but…”. This format differs sharply from NYCC’s single-panel, narrative-incongruity humor. To clarify, the YesBut benchmark contains four subtasks with different formats, but only two of them, Underlying Philosophy Selection (“Philosophy”) and Title Matching (“Title”), are classification-based and therefore directly comparable to NYCC-style caption selection. We focus on these two because they probe the same core ability as NYCC: interpreting a visual setup, identifying the source of humor or contrast, and selecting the best textual option among competitors. The strong gains we observe on these tasks indicate that the captionist-style reasoning learned from NYCC generalizes beyond the domain it was trained on.
>
> ### YesBut (NeurIPS 2024)
>
> | Model         | Philosophy ↑        | Title ↑           |
> |---------------|----------------------|--------------------|
> | Qwen2.5-VL-7B-Instruct (Base) | 43.19              | 29.11             |
> | **Ours-7B**   | **74.90**     | **63.32**   |
>
> These results, **\+32% on Philosophy and \+34% on Title**, show that the model learns *transferable reasoning patterns*, not contest-specific memorization. In particular, the captionist-style reasoning traces (which decompose a scene into incongruity → interpretation → punchline) appear to generalize naturally to the contrastive logic required in YesBut.
>
> #### **(b) Generalization to broader vision-language semantics: DeepEval (humor subset)**
>
> DeepEval is not a humor dataset; the humor subset constitutes only **29 samples (2.9% of the benchmark)**. It tests semantic grounding, descriptive accuracy, and title selection—tasks that probe understanding but not NYCC-style humor.
>
> Again, our 7B model substantially outperforms its base counterpart:
>
> ### DeepEval (Humor Subset)
>
> | Model                         | DeepSemantics ↑ | Description ↑ | Title ↑ |
> |-------------------------------|-------------|----------------|----------|
> | Qwen2.5-VL-7B-Instruct (Base)  | 10.34       | 24.13          | 34.48    |
> | **Ours-7B**        | **54.43**   | **100.00**     | **63.18** |
>
> Despite the limited humor content in DeepEval, our model improves semantic judgments and descriptive grounding by a significant margin. This confirms that the reasoning supervision improves general multimodal interpretation, not simply humor classification.
>
> Hu et al., Cracking the Code of Juxtaposition: Can AI Models Understand the Humorous Contradictions, NeurIPS 2024\.
>
> Yang et al., Can Large Multimodal Models Uncover Deep Semantics Behind Images? Findings of ACL 2024\.

---

> ### Author Response · Authors · 2025-11-21
>
> **3\. “LLM-as-judge may introduce bias. How do you justify correctness?”**
>
> Both perception and style rewards are computed by prompting the judge LLM with *structured checklists* derived from human-defined criteria (e.g., grounding in salient entities; adherence to captionist stylistic conventions such as everyday phrasing, wordplay, and punchline placement). The judge LLM does not “decide what is funny”; it only verifies whether the model’s reasoning satisfies these pre-specified conditions. This greatly limits opportunities for bias.
>
> As an empirical validation, we explicitly checked whether the judge assigns higher auxiliary rewards to *correct* predictions. Concretely, we analyzed all cases where either perception or style reward exceeded 80% of their observed maximum.
> **Over 70% of these high-reward cases correspond to correct final answers**, indicating that the reward signal is not arbitrary: the rubric-guided judge reliably provides higher scores when the model selects the correct caption.
>
> This empirical correlation demonstrates that the LLM-as-judge component functions as a *useful, complementary guidance signal*, reinforcing the intended reasoning behaviors rather than introducing noise.
>
> **4\. “Can more cognitive or cultural theory be integrated?”**
> Thank you for this helpful suggestion. We agree that humor is fundamentally a cognitive and cultural phenomenon, and we have updated the manuscript to make these connections explicit.
>
> **(a) Our reasoning supervision directly encodes core cognitive theories of humor.**
> The structure of our captionist-style reasoning traces is grounded in classic cognitive theories, **incongruity identification → reinterpretation → resolution**, as formalized in incongruity–resolution models (Suls, 1972), script opposition (Attardo & Raskin, 1991), and frame-shifting accounts (Coulson, 2001). These stages are not incidental: they form the explicit template for every supervised reasoning trace.
>
> **(b) Our RL rewards operationalize perceptual and cultural dimensions of humor.**
> The **perception reward** enforces grounding in visually salient incongruities (a cognitive prerequisite for humor understanding), while the **style reward** captures cultural and editorial norms emphasized in professional caption writing, lexical economy, idiomatic phrasing, metaphorical compression, and punchline placement. These criteria come from documented captionist practice rather than generic LLM heuristics.
>
> **(c) Expert feedback informed the design philosophy.**
> During development, we collaborated with an established NYCC expert, who emphasized that human humor comprehension relies on layered reasoning about narrative roles, visual conflict, and culturally calibrated wit. This perspective directly shaped both our reasoning-trace format and reward design.
>
> In summary, while the training stages (CPT→SFT→RL) are standard, **the supervision content and reward structure are tightly grounded in cognitive and cultural theories of humor**, and we have revised the Approach sections to highlight these connections more clearly.
>
> **5\. “Can you improve human evaluation? One expert is insufficient.”**
> Thank you for pointing this out. Humor is inherently subjective, and we agree that a single expert cannot fully represent cultural or linguistic variation. In response, we conducted a **broader user study with 21 participants** from varied cultural backgrounds, covering a subset of evaluation questions. These results are now included in the revised version and provide a more diverse human baseline.
>
> In addition, we are preparing qualitative comparisons between model-generated reasoning and expert reasoning (to be included in the Appendix) to highlight where VLMs align with, or diverge from, human interpretive strategies.
>
> **6\. Typos and formatting**
> All noted issues have been fixed in the revised version.

---

### Official Review · Reviewer_93PD · 2025-10-30

**Soundness:** 2
**Presentation:** 3
**Contribution:** 2
**Rating:** 2
**Confidence:** 4

**Summary:**

This paper investigates the task of multimodal humor understanding and introduces a training framework designed to enhance the reasoning capabilities of MLLMs in interpreting cartoons. The proposed framework comprises three stages: (1) continued pretraining on humor-related corpora for domain adaptation; (2) supervised fine-tuning (SFT) using synthetic reasoning traces generated by DeepSeek-R1 and GPT-4o to cultivate reasoning ability; and (3) GRPO-based reinforcement learning with grounded perception and stylistic rewards to further ehance the model’s humor reasoning. Experiments conducted on two humor understanding benchmarks (both requiring the model to match captions with corresponding cartoon images) demonstrate that the proposed method significantly outperforms several strong MLLM baselines.

**Strengths:**

- This paper investigates an important task of humor understanding. I believe this an essential direction, as the ability of current large models to comprehend human creative expressions remains underexplored yet highly significant.
- The proposed framework is well-motivated and systematically designed, incorporating continued pretraining, supervised fine-tuning, and RL stages. Experimental results demonstrate the model’s strong performance across two humor understanding benchmarks.
- Overall, the paper is clearly written, well-organized, and easy to follow.

**Weaknesses:**

- My main concern lies in the novelty of the paper. The proposed training framework largely follows a conventional large-model training pipeline, comprising continued pretraining, supervised fine-tuning, and reinforcement learning, and primarily applies it to a new domain, namely humor understanding. As a result, the overall contribution feels somewhat incremental rather than fundamentally novel;
- Regarding continued pretraining, Table A.1 shows that the dataset contains only 4,123 instances (fewer than 5 million words). Fine-tuning on such a small dataset may risk overfitting to this specific domain, leading to limited generalization. Moreover, the pretraining corpus appears highly similar to the NYT Cartoon dataset, which closely overlaps with the evaluation benchmark (e.g., the NYT Cartoon Benchmark). This overlap raises concerns about potential data leakage or domain bias. Additionally, humor understanding encompasses diverse subdomains (e.g., various comic genres as in [1,2]); hence, the proposed pretraining approach may not generalize well to humor datasets with significant domain shifts.
- Concerning reinforcement learning, the motivation for including the style reward ($R_s$) is unclear. Since the task involves matching an image with an existing caption rather than generating the caption itself, it is not evident how stylistic rewards contribute to improving the model’s understanding of humor or visual-semantic alignment. On the contrary, such rewards might introduce additional inductive biases.
- The benchmark coverage is limited. The evaluation focuses only on two datasets, both closely related to NYT-style cartoons. It would strengthen the work to assess performance on more diverse humor understanding benchmarks, such as [1] and [2].


[1] Cracking the Code of Juxtaposition: Can AI Models Understand the Humorous Contradictions, NeurIPS 2024

[2] Can Large Multimodal Models Uncover Deep Semantics Behind Images? ACL Findings 2024

**Questions:**

Q1. For continue pretraining: As the corpora are textual data, how do you conduct model training of MLLMs? Do you freeze the ViT and only update the language model component?

Q2. Missing Citations on some recent humor understanding works, such as [1] and [2] mentioned in the weaknesses.

---

> ### Author Response · Authors · 2025-11-21
>
> We thank the reviewer for the detailed and constructive feedback. We are glad that the reviewer finds the paper well-motivated, clearly written, and addressing an important but underexplored problem. Below, we respond to each concern in turn.
>
> **1\. “The contribution feels incremental; the pipeline resembles a standard CPT→SFT→RL recipe.”**
> Thank you for raising this important concern. While our framework follows the widely used CPT → SFT → RL structure, the novelty of our work does **not** lie in proposing a new optimization algorithm, but in **adapting these components to a domain where existing multimodal pipelines fundamentally fail**.
>
> Humor understanding, especially New Yorker–style visual-verbal incongruity, requires a type of reasoning that differs sharply from standard VQA, captioning, or preference prediction. Professional captionists analyze cartoons through a multi-step interpretive workflow:
>
> 1. reconstruct the visual narrative and character roles,
> 2. identify the incongruity driving the joke,
> 3. explore alternative narrative resolutions,
> 4. evaluate which punchline resolves that incongruity in the most surprising way.
>
> This workflow is documented extensively by editors and contest veterans.
>
> Our contribution is to **operationalize this human expert process inside a multimodal model**, via three domain-specific mechanisms:
>
> * **Captionist-style reasoning traces** that encode the exact steps used by professional captionists;
> * **Humor-specific RL rewards**, grounded in visual anchoring and editorial stylistic principles, which are core to real cartoon judging;
> * **CPT on humor discourse**, shifting the representation space so that later reasoning and RL actually take hold (which does *not* happen when starting from a generic multimodal backbone).
>
> This combination reframes humor understanding from a shallow caption-matching problem into a **transparent, interpretable reasoning task**—to our knowledge, the first structured attempt to do so.
>
> We also note that humor understanding, especially New Yorker–style single-panel cartoon humor, is not a trivial pattern-recognition task but a complex human cognitive skill. Professional captionists require years of practice to master the workflow of noticing narrative roles, decomposing visual incongruity, and crafting an interpretive resolution. In discussions with a highly established captionist with decades of experience judging the contest, who has judged hundreds of contests and collaborated with us in this project, the expert emphasized that this reasoning process is difficult even for humans to articulate, let alone teach to a model. From their perspective, the scientific contribution lies precisely in demonstrating that comparatively small open models (7B–32B), guided by captionist-style reasoning traces and humor-aware RL, can approximate this expert workflow. This ability goes beyond standard fine-tuning pipelines and addresses a domain where frontier models themselves often fall short.
>
> In summary, while our architecture follows standard stages, our novelty lies in **what those stages are teaching**: a formalized, expert-grounded reasoning process tailored to visual-verbal humor. This represents a principled reframing of the humor-understanding problem and yields measurable, scalable improvements in multimodal reasoning.

---

> ### Author Response · Authors · 2025-11-21
>
> **2\. “CPT dataset is small; risk of overfitting; possible overlap with NYCC evaluation data.”**
> We agree that CPT on small corpora may raise concerns about overfitting and domain overlap. As documented in the supplementary (Appendix A), to mitigate this: (i) We also incorporate **general-purpose data**, subsets from FineWeb (Penedo et al., 2024\) and OLMo-Mix-1124 (AllenAI, 2024), to prevent catastrophic forgetting and avoid overspecialization. (ii) We checked for similarity-based overlap with the NYCC evaluation sets using CLIP embeddings for images and additionally conducted manual inspection. This ensures that no cartoons or captions in our CPT corpus appear in any evaluation benchmark. (iii) Our CPT corpus is stylistically related but not semantically overlapping with the evaluation data. The goal is domain adaptation, not memorization.
>
> Regarding generalization beyond NYCC-style content; in the revised submission, we include **zero-shot evaluations** on additional humor benchmarks, suggested by the reviewer, and the model continues to improve over its base variant, indicating that it does not overfit to NYCC-specific distributions.
>
> **3\. “Style reward seems irrelevant to matching tasks.”**
> We appreciate the opportunity to clarify this. Although the task involves selecting a caption, the justification step is crucial:
>
> * Many NYCC captions differ only subtly in tone, idiom, or punchline timing.
> * Professional captionists (Mankoff 2014, Wood, 2024\) explicitly judge captions by *linguistic execution*—not just incongruity fit.
> * The “style” reward **does not encourage generating stylized captions**; it evaluates whether the *reasoning explanation* references the linguistic properties that differentiate strong vs. weak captions.
>
> This matters because:
>
> **Matching/ranking correctness often hinges on explaining why one caption *reads* funnier than another.** For example, choosing a caption that misuses idiom/pacing is often wrong even if it references the correct visual object.
>
> The ablation table confirms this: adding the style reward improves ranking accuracy, particularly on fine-grained humor tasks.
>
> Thus, the style reward is not an extraneous inductive bias. It encodes *captionist linguistic discriminators* that are decisive for humor understanding.

---

> ### Author Response · Authors · 2025-11-21
>
> **4\. “Benchmark coverage limited; evaluate on more diverse humor datasets.”**
> Thank you for this suggestion. We agree that evaluating beyond NYCC is important, especially because humor varies widely across formats, cultures, and cognitive structures.
>
> Following your recommendation, we expanded our experiments in the revised submission to include **both datasets you cited**, *YesBut* (NeurIPS 2024\) and the *DeepEval Humor Subset* (ACL 2024). Importantly, we test our model trained **exclusively on NYCC data** in a *zero-shot* setting on these external benchmarks, so the results directly measure cross-domain generalization.
>
> ### **(a) Contrastive/juxtaposition visual humor: YesBut (NeurIPS 2024\)**
>
> YesBut evaluates humor through **two-panel, contrastive juxtapositions** (“yes… but…”), a structure that differs fundamentally from NYCC’s single-panel incongruity–resolution format.
>
> YesBut contains four subtasks, but **only two of them**, namely Underlying Philosophy Selection (“Philosophy”) and Title Matching (“Title”), match NYCC’s classification setting, where the model must choose the correct caption among competitors. These are therefore the most meaningful tasks for a comparison.
>
> Our NYCC-trained model achieves large, consistent improvements over the base model:
>
> ### YesBut (NeurIPS 2024\)
>
> | Model         | Philosophy ↑        | Title ↑           |
> |---------------|----------------------|--------------------|
> | Qwen2.5-VL-7B-Instruct (Base) | 43.19              | 29.11             |
> | **Ours-7B**   | **74.90**     | **63.32**   |
>
> These gains (+32% on Philosophy, \+34% on Title) indicate that the captionist-style reasoning we teach—*identify incongruity → interpret narrative → choose the best textual resolution*—generalizes naturally to the contrastive logic in YesBut. This goes well beyond the stylistic domain of NYCC.

---

> ### Author Response · Authors · 2025-11-21
>
> ### **(b) Broader multimodal semantics: DeepEval (ACL 2024, Humor Subset)**
>
> DeepEval is a general semantic vision–language benchmark, not a humor dataset; only **29 images (2.9% of the dataset)** belong to the humor subset. It tests:
>
> * **semantic alignment**,
> * **descriptive grounding**, and
> * **title selection**,
>
> rather than NYCC-like punchline reasoning.
>
> Despite this domain shift, our 7B model again substantially improves over the base model:
>
> ### DeepEval (Humor Subset)
>
> | Model                         | DeepSemantics ↑ | Description ↑ | Title ↑ |
> |-------------------------------|-------------|----------------|----------|
> | Qwen2.5-VL-7B-Instruct (Base)  | 10.34       | 24.13          | 34.48    |
> | **Ours-7B**        | **54.43**   | **100.00**     | **63.18** |
>
> Even though DeepEval contains minimal humor content, the trained model improves semantic and descriptive reasoning by a wide margin, showing that the learned reasoning structure enhances **general multimodal understanding**, not only humor-specific skills.
>
> These findings demonstrate that the framework does not overfit to New Yorker–style humor. Instead, it captures a transferable reasoning schema (incongruity → interpretation → evaluation) that applies across diverse humor forms and even outside humor entirely.
>
> We have also added these results to the supplementary of the revised manuscript.
>
> **Q1.“How do you train the MLLM during CPT with text-only data? Do you freeze the vision tower?”**
> Yes. During CPT, we freeze the entire vision encoder and update only the language model and cross-modal fusion layers. This is now stated explicitly in the revised version of the supplementary material..
>
> **Q2. “Missing citations”**
>
> Thank you for pointing this out. We have added citations to:
>
> * Hu et al., *Cracking the Code of Juxtaposition* (NeurIPS 2024\)
> * Yang et al., *DeepEval* (ACL Findings 2024\)
>
> and expanded the corresponding related work section.

---

> > ### Comment · Reviewer_93PD · 2025-11-28
> > **Response to Rebuttal**
> >
> > Thanks for your detailed rebuttal, which has addressed some of my concerns, including CPT data generalization, domain adaptation to other comics, and training details. Based on these clarifications, I would increase my overall rating and soundness score.
> >
> > In addition, I still have a few points I would like to further confirm:
> >
> > - On the necessity of CPT: I remain uncertain about whether this stage is essential, given that pretrained models have already been exposed to extensive textual corpora. Although the authors emphasize the inclusion of general-purpose data, I am not fully convinced that domain adaptation on stylistically related corpora is the key factor for improving humor understanding. Humor comprehension depends more on underlying social and cultural knowledge rather than stylistic adaptation to a specific domain.
> >
> > - Ablation on SFT and RL without CPT: Related to the above point, I wonder whether you have evaluated the individual effects of SFT and RL when used without the CPT stage, i.e., BASE + SFT and BASE + RL. Since Qwen2.5-VL-Instruct is already a strong instruction-tuned model, including these variants would more clearly illustrate the contribution of each stage and validate whether CPT brings additional benefits beyond what the base model is already capable of.
> >
> > - While large language models already perform well on linguistic reasoning, comic understanding also hinges on visual interpretation, which is a weakness of current large models. Given this, I believe that enhancing the visual perception component would be more impactful than CPT in terms of improving model capability and strengthening the novelty of the paper.

---

> ### Author Response · Authors · 2025-12-03
>
> ### **1\. On the necessity of CPT**
>
> Thank you for raising this important question. We agree that CPT should be justified not as *“extra fine-tuning”* but as a mechanism that provides the model with **the right inductive priors before reasoning supervision begins**.
>
> What CPT contributes is not stylistic imitation, but **contextual grounding** for the reasoning traces used in SFT. Our SFT traces assume familiarity with:
>
> * how readers interpret *narrative roles* in cartoons,
> * how incongruity is typically resolved in editorial humor,
> * which cultural assumptions captionists rely on when selecting a punchline,
> * how a scene is mentally parsed before a caption is chosen.
>
> In fact, the CPT corpus, composed of judging discussions, roundtables, and craft-focused books, offers **explanations of these mental steps**, not just examples of captions. Without CPT, SFT must teach both the *procedure* and the *background assumptions* simultaneously. With CPT, SFT can focus on teaching **structure**, not **worldbuilding**.
>
> This is precisely reflected in the new ablations (Table G.1):
>
> * **BASE \+ SFT** improves performance, but
> * **CPT \+ SFT** consistently improves further,
>   indicating that CPT provides *missing prerequisites* that the base model lacks.
>
> In short, CPT is not used to “specialize the model to New Yorker style” but to **prepare the latent space so that captionist-style reasoning is learnable at all**.
>
> ---
>
> ### **2\. On SFT and RL without CPT (new ablations)**
>
> Thank you for this suggestion. We have added new ablations isolating the effects of SFT and RL without CPT (Table G.1 in the revised appendix). These results clarify the distinct role of each stage:
>
> **(a) SFT provides the core gains in structured reasoning.**
> Base \+ SFT yields large improvements across all tasks, confirming that the captionist-style reasoning traces, rather than CPT, are the primary driver of multi-step analytic behavior. This aligns with our goal: SFT teaches the model *how* captionists reason.
>
> **(b) CPT specifically enhances SFT, but not RL applied in isolation.**
> Empirically, Base \+ CPT \+ SFT consistently outperforms Base \+ SFT, showing that CPT provides *background conceptual grounding*, editorial heuristics, narrative expectations, and incongruity-resolution patterns, that SFT explicitly builds upon.
>
> However, CPT does **not** help RL when SFT is absent. In fact, Base \+ RL slightly outperforms Base \+ CPT \+ RL. This matches well-understood dynamics in RLHF: RL optimizes task-level reward signals (correctness, grounding, formatting), which do not directly leverage the conceptual priors introduced by CPT. Moreover, CPT broadens stylistic and narrative variability in the initial policy, producing a small distribution shift that makes RL optimization harder when no structured SFT step precedes it.
>
> **(c) The three stages are complementary, with the full pipeline performing best.**
> The strongest results arise from the full pipeline (CPT \+ SFT \+ RL). CPT supplies the cultural and editorial context assumed by captionist-style traces; SFT teaches the structured analytic process; and RL reinforces correctness and grounding. The ablations therefore reflect a coherent division of labor across training stages.
>
> ---
>
> ### **3\. On the importance of visual perception**
>
> We agree with the reviewer’s core intuition: the main bottleneck in cartoon understanding is the **vision encoder**, not the language model. Modern MLLMs perform well at linguistic reasoning, but stylized cartoons challenge current vision backbones because they require:
>
> * recognizing caricatured shapes,
> * disambiguating symbolic props (e.g., scythes vs. swords),
> * interpreting visually implied narratives.
>
> This is precisely why we introduced the **perception reward**—to compensate for the encoder’s weakness via additional grounding supervision. While this cannot fully replace a stronger vision backbone, it yields measurable improvements on ranking tasks.
>
> We now explicitly document this in Appendix E where we discuss visual perception errors that **real progress on the visual side will likely require larger, publicly available cartoon corpora**, which currently do not exist at scale.

---

### Official Review · Reviewer_fKK3 · 2025-10-31

**Soundness:** 3
**Presentation:** 3
**Contribution:** 3
**Rating:** 4
**Confidence:** 3

**Summary:**

This paper does something interesting: instead of just having AI guess what's funny, it tries to teach it to think like a New Yorker captionist. The main contribution is shifting humour from a black-box guessing game to an interpretable, step-by-step reasoning task. The technical pipeline is typical and standard.

**Strengths:**

1. The paper doesn't just stop at which is funnier? It originally proposes teaching AI to mimic the human thought process.
2. This paper isn't just about jokes; it's about explainable AI and tackling other creative tasks that need cultural context and subjective judgment.
3. The technical pipeline is standard. The two RL rewards ($R_p$ and $R_s$) are especially clever. They quantify a fuzzy concept like good humour into optimizable signals for visual grounding and stylistic taste. The ablation studies (Tables 2 and 3) are well-executed and clearly demonstrate the contribution of each part.
4. The paper is very clearly written. The introduction and abstract are concise, and the figures provide a clear overview. The authors are also very candid about the limitations in the Appendix.

**Weaknesses:**

1. The SFT traces are not from human experts, but are synthetically generated by GPT-4o. This means the model is merely distilling one AI's (potentially flawed) reasoning, rather than learning from genuine human captionists. The RL reward signal is also not from human preference, but from an LLM-as-judge. This creates a closed loop where an AI is trained on an AI's data and judged by another AI, with no validation of whether this aligns with genuine human humour.

2. For all its complexity, the final results in Table 1 are weak. The proposed model (60.33 on Matching) is dramatically outperformed by the closed o3 baseline (83.33). This >20-point gap calls into question the practical utility of this pipeline.

3. The paper's claim to align with expert humour is severely undermined by its evaluation. The only human baseline mentioned is a single Human expert. This N=1 evaluation is statistically meaningless and lacks diversity. Judging something as subjective and culturally specific as New Yorker humour requires, at the very least, a diverse panel of human evaluators, not a single data point.

**Questions:**

1. The SFT stage relies entirely on teacher traces generated by GPT-4o. Was any human-led quality control or evaluation performed on these synthetic traces? How do you know that GPT-4o's generated reasoning is high-quality and free of plausible-sounding nonsense?
2. You benchmark against only one professional captionist. Could you consider including more evaluators? Additionally, a deeper quantitative or qualitative analysis is needed, for instance, to examine the differences between human and VLM humour comprehension and to identify which specific aspects of humour VLMs fail to grasp.
3. Why did your Perception Reward ($R_p$) fail to correct the fundamental Viking/Grim Reaper error? Does this mean your $R_p$—which relies on text anchors—can only check for text-to-text consistency, not fix actual visual perception errors?
4. Is it possible to experiment on more of the latest 7B models to see if this could bring different potential improvements?

---

> ### Author Response · Authors · 2025-11-21
>
> We thank the reviewer for the thoughtful and constructive assessment. We are pleased that the reviewer finds the paper clearly written, the reasoning-based framing original, and the RL reward design meaningful. Below we address each concern in detail.
>
> **1\. “SFT traces are synthetic; does the model merely distill flawed AI reasoning?”**
> We appreciate this concern and clarify that the SFT traces are not unconstrained LLM outputs. They follow **a structured reasoning template derived from real captionist practice**, grounded in authoritative captionist methodologies documented by Mankoff (2002, 2014\) and Wood (2024). These sources document a consistent workflow used by professional New Yorker captionists—scene reconstruction → incongruity detection → narrative reinterpretation → punchline evaluation. Our SFT prompting template **explicitly encodes this human-derived structure**, ensuring the resulting traces instantiate a documented captionist thought process rather than free-form LLM intuition.
>
> We also manually inspected a diverse subset of traces and found them structurally consistent, correctly grounded in the scene, and aligned with expert heuristics. Their purpose is to teach the model *how to decompose a cartoon*, not to replicate humor preferences. This use of structured teacher traces is consistent with recent findings (Muennighoff et al., 2025\) showing that the value of SFT lies in the reasoning *template*, even when occasional details may be imperfect.
>
> Thus, the traces reflect **human-grounded reasoning patterns**, not circular AI-only supervision.
>
> \* Muennighoff et al., s1: Simple test-time scaling,  ICLR 2025 Workshop on Reasoning and Planning for LLMs, 2025
>
> ***2\.*** **“Human evaluation is N=1 and insufficient for a subjective domain like humor.”**
> We appreciate this feedback. For this revision, we conducted a 21-participant user study covering a subset of matching and ranking items, providing a population-level human baseline that complements the expert’s more editorial perspective. These results are now included in the main comparison table (as “Human (Non-Expert)”).
>
> Together, the expert baseline reflects captionist-style reasoning, and the participant baseline reflects crowd-level humor preferences. This directly addresses the reviewer’s concern and provides a more complete human reference point.
>
> Additionally, we are preparing qualitative comparisons between model-generated reasoning and expert reasoning (will be included in the Appendix) to illustrate where VLMs diverge from human intuition.

---

> > ### Author Response · Authors · 2025-11-26
> >
> > Dear Reviewer fKK3,
> >
> > As promised in our rebuttal, we have now added a qualitative comparison between model-generated reasoning and expert reasoning in Appendix D of the updated manuscript (Appendix D.4). This supplement illustrates how the model adopts the structured analytic process used by expert captionists and provides additional insight into the interpretability of its reasoning traces.
> >
> > We hope this clarification is helpful and further strengthens the contribution. Please let us know if there is any additional analysis that would be helpful during the discussion phase.
> >
> > Thank you again for your thoughtful and constructive review.

---

> ### Author Response · Authors · 2025-11-21
>
> **3.“The model is far below o3 on Matching; is the pipeline practically useful?”**
> This is an important question. We highlight two key clarifications:
>
> **(i) Closed models have structural advantages on Matching.**
> Matching requires selecting the exact winning caption, which naturally favors models with access to large proprietary corpora. For example, OpenAI has publicly announced a partnership with Condé Nast (see: *OpenAI–Condé Nast collaboration announcement, 2024, [https://openai.com/index/conde-nast/](https://openai.com/index/conde-nast/)*). While we make no claim about direct training exposure, such agreements provide access to extensive New Yorker material and create inherent asymmetries. Winner-identification tasks therefore reward recall of domain-specific textual patterns more than genuine humor reasoning.
>
> **(ii) Where memorization cannot help (ranking settings), our model is strongest.**
> All three ranking evaluation setups, (Hessel et al., 2023\) Ranking, 10-vs-1000, and 30-vs-300, require resolving subtle humorous alternatives rather than recalling a single known winner. Closed models cannot rely on memorization or domain overlap in these settings. Across these ranking tasks, our 32B model consistently achieves the strongest performance among multimodal models, surpassing o3 on the primary Hessel et al.’s Ranking split and closely approaching it on the more challenging wide-gap and mid-gap variants. This indicates that our reasoning-trace supervision and humor-aware RL improve **genuine humor understanding**, not dataset familiarity.
>
> **(iii) Scaling experiments show the pipeline’s practical value.**
> Applying our framework to the larger 32B backbone substantially improves performance across *all* ranking tasks and narrows the gap on Matching, despite the lack of access to any proprietary or privileged corpora. This scaling behavior demonstrates that the proposed method is effective, robust, and beneficial for open models.
>
> **4\. “Why did the Perception Reward fail to fix the Viking/Grim Reaper error?”**
> The reviewer is correct that Perception Reward cannot correct all perception errors. It rewards *alignment between reasoning and salient scene anchors*, but it cannot compensate for fundamental misrecognitions in the underlying vision encoder, especially for stylized line-art cartoons.
>
> Crucially, with the 32B backbone, our CPT+SFT+RL model **correctly interprets the Viking scene**, indicating that the limitation stems from backbone capacity and pretraining coverage, not the reward design itself. We now explicitly note this (with illustrative examples) in the supplemental.
>
> **5\. “Could you test more modern 7B models?”**
> Our pipeline is architecture-agnostic and compatible with newer 7B models. Given compute constraints, we prioritized evaluating a significantly larger 32B model, which provides a more informative scaling signal and shows clear benefits of our method. We now emphasize this in the revision and list expanded 7B evaluation as future work.

---

### Official Review · Reviewer_QC7F · 2025-11-01

**Soundness:** 3
**Presentation:** 3
**Contribution:** 2
**Rating:** 4
**Confidence:** 3

**Summary:**

This paper proposes a three-stage framework. First, continual pretraining guides the model’s representation space toward a specific sense of humor by feeding it curated humorous corpora. Second, the authors perform supervised fine-tuning using a constructed reasoning traces dataset to encourage captionist-style reasoning in the model. Finally, they apply reinforcement learning, introducing two additional rewards—visual perception and style—to further enhance the model’s ability to discriminate humor. This framework shifts the large language model from merely understanding humor through surface patterns to truly engaging in creative reasoning.

**Strengths:**

- The three-stage framework is both novel and well-designed, effectively mimicking the reasoning patterns of human captionists, thereby enabling multimodal large models to acquire similar capabilities.
- The reasoning trace dataset constructed by the authors, along with the visual reference used for reinforcement learning, played a significant role in improving the model, making it a substantial contribution to the work.

**Weaknesses:**

- Typo in line 40 "thrieves"
- The motivation is not clearly stated. Why is fine-tuning necessary? (See Questions.)
- The experimental results do not appear to be the best. (See Questions.)

**Questions:**

- The paper uses a large language model to construct both the reasoning trace dataset and the visual reference. Since the model already demonstrates the ability to generate high-quality humorous captions through carefully designed prompts, why is fine-tuning still necessary? It seems that simply designing a method to guide the model would already be sufficient.
- As shown from the experimental results, the proposed method does not achieve the best performance. Since some closed-source models like o3 already perform well enough, the motivation of this work appears less compelling. Could the authors provide further clarification on this point?
- A minor suggestion is, you could use special markers or colors to help readers quickly grasp the key points in the presented experimental results, such as Table 1.

---

> ### Author Response · Authors · 2025-11-21
>
> We thank the reviewer for the constructive feedback and are encouraged by the positive assessment of our three-stage framework, the design of our reasoning-trace dataset, and the overall alignment of our approach with human captionist reasoning. We address the reviewer’s questions regarding fine-tuning motivation and model comparisons in detail below.
>
> **Q1. “Why is fine-tuning necessary if an LLM can already generate humorous captions with prompting?”**
> Thank you for raising this point. We believe the concern stems from a misunderstanding of what our model is trained to do and how our data are constructed.
>
> **(1) Our models do not generate humorous captions. They perform humor understanding.**
> All tasks in our benchmark, matching and ranking, operate exclusively over *existing* human captions. The SFT reasoning traces are not humorous generations but **explanations** of why one caption is funnier or more aligned with the cartoon. Their purpose is to teach the model *how* expert captionists reason, not to produce humor.
>
> **(2) The reasoning traces come from DeepSeek-R1 using human-annotated scene descriptions.**
> DeepSeek-R1 does *not* create novel humor. It produces structured explanations conditioned on *human captions* and human-provided scene annotations, closely mirroring professional captionist reasoning. Thus, the SFT dataset encodes **human-aligned reasoning structure**, not model-generated creativity.
>
> **(3) Base open-weight MLLMs lack the multimodal abstraction required for humor reasoning.**
> Unlike large proprietary systems, Qwen2.5-VL-7B (and the newly added Qwen2.5-VL-32B) do not exhibit strong humor reasoning or robust instruction-following out of the box. They struggle with identifying visual incongruities, linking them to cultural knowledge, evaluating captionist logic, and producing coherent explanations.
>
> Our experiments show clear gaps between the **base models** and the **CPT/SFT/RL-trained versions**, demonstrating that fine-tuning is essential for transferring captionist-style visual–semantic reasoning into these smaller open models.
>
> **(4) Guiding a large proprietary model via prompting is not the goal of this paper.**
> Our goal is to equip *small, open, reproducible multimodal models* with interpretable humor reasoning skills. Prompt engineering a frontier closed-source model (e.g., o3) would not advance transparent research or model interpretability, and would not produce models that can be deployed or studied openly.
>
> In summary, fine-tuning is necessary because the goal is not to elicit funnier captions from a large proprietary LLM, but to **teach open models a structured captionist reasoning process that they fundamentally lack.** This is essential for developing reproducible and interpretable multimodal reasoning systems in an underexplored, human-centric domain like humor.

---

> ### Author Response · Authors · 2025-11-21
>
> **Q2. “Why is this work compelling if closed-source systems like o3 perform well?”**
> We appreciate this concern. Our motivation is **not** to “beat o3 at all costs,” but to advance transparent, reproducible multimodal reasoning in a domain where open models currently lag far behind.
>
> **(1) Closed models benefit from scale and unknown proprietary data.**
> While OpenAI has not disclosed the exact parameter count of o3, independent technical analyses suggest that it likely operates in the large-model regime (hundreds of billions of parameters), giving it substantial capacity advantages over open 7B–32B models. Given publicly known licensing partnerships (e.g., Condé Nast–OpenAI, [https://openai.com/index/conde-nast/](https://openai.com/index/conde-nast/)), it is plausible that models such as o3 have been exposed to New Yorker Caption Contest material or large humor-related corpora unavailable to the community.
>
> Despite this structural disadvantage, our approach enables open models to close the gap meaningfully. Following the reviewer’s suggestion, we also evaluated a stronger 32B backbone. As shown in Table 1 (updated), our 32B model now attains **state-of-the-art performance on the main ranking task** (Hessel et al., 2023), outperforming o3, and delivers the **second-best results** on the remaining ranking tasks. For convenience, we reproduce the key excerpt here:
>
> **Updated Key Results (Table 1 excerpt)**  **Accuracy (%) on caption matching and ranking tasks.**
> | Model                 | Matching | Ranking | 10-vs-1000 | 30-vs-300 |
> |----------------------|----------|---------|------------|-----------|
> | **o3**               | **83.33** | 62.85  | **69.05**  | **54.57** |
> | Qwen2.5-VL-32B       | 46.67    | 49.87  | 52.00      | 44.57     |
> | **Ours (32B)**       | 62.67    | **68.05** | 62.86     | 53.14     |
>
> We note that the *matching* task is inherently different: it requires selecting the exact winning caption among a very small set, which strongly benefits models that may have been exposed (directly or indirectly) to New Yorker Caption Contest material or similar paywalled sources. In contrast, the *ranking* tasks measure a model’s ability to **reason about incongruity, resolve competing humorous interpretations, and justify preference decisions**, skills that cannot be reduced to memorization.
>
> The fact that our open 32B model surpasses or closely approaches closed systems on these reasoning-heavy tasks demonstrates that the proposed training pipeline effectively imparts high-level humor understanding. But we acknowledge that the matching task still leaves room for improvement, and we explicitly discuss these limits and potential future directions in the updated submission.
>
> **(2) Our scientific contribution lies in enabling open models to reason about humor.**
> Humor understanding, especially visual-verbal incongruity resolution, is one of the least studied areas in multimodal reasoning. Our pipeline contributes **a transparent reasoning-trace dataset**, **a reproducible 3-stage training pipeline**, improved **humor-aware multimodal reasoning in 7B/32B models**, and **interpretable, traceable model behaviour**. Closed systems cannot offer these benefits:  their training data, alignment procedures, and internal behaviors are opaque.
>
> **(3) Improving small open models is essential for research, fairness, and real-world deployment.**
> If progress were judged solely by outperforming very large proprietary models, open and reproducible research would stall. Our results demonstrate that (i) **expert reasoning can be transferred,** (ii) **significant gains are achievable at small–medium scales, and** (iii) **open models can approach closed-source performance through principled training.**
>
> In summary, our work provides a transparent, reproducible path for endowing open multimodal models with captionist-style reasoning. This remains impactful regardless of whether certain closed systems report higher raw accuracy.
>
> We hope these clarifications resolve the reviewer’s concerns and make clear that our submission offers a principled and reproducible path toward advancing open multimodal humor reasoning.
>
> **On Improving clarity of experimental tables**
> Thank you for the suggestion. In the revised submission, we highlight key results through clearer formatting (e.g., boldface and improved layout) so readers can more easily compare models across tasks.

---

### Author Response · Authors · 2025-11-26

Dear Reviewers and AC,

We sincerely thank all reviewers for their thoughtful and constructive evaluations. Across the reviews, several strengths were consistently highlighted, including the clarity of the paper, the motivation for modeling humor as a structured reasoning process, and the consistent empirical gains across multiple benchmarks. We greatly appreciate this careful and positive assessment.

Following the discussion phase, we previously submitted a revised manuscript addressing all reviewer concerns in depth, including expanded analyses, additional human-subject results, and clarifications of our training pipeline.

As a final minor update, as per Reviewer fKK3's request, we have now added qualitative comparisons between model-generated reasoning and expert reasoning in Appendix D (D.4). These illustrate how the model’s analytic structure aligns with expert strategies even when their final choices differ.

We thank the reviewers once again for their time and engagement, and we hope that the revisions fully address the remaining questions.

Authors of Submission #9836

---

### Author Response · Authors · 2025-12-03
**Summary of Rebuttal & Revisions for Paper #9836 (Humor Understanding)**

Dear Area Chair,

Due to the recent changes in the review process, we understand you are likely assessing our submission, for the first time. To assist you, we provide a concise summary of how we successfully addressed the primary reviewer concerns in our revised manuscript.

**Engagement & Score Status:** Before the discussion freeze, **Reviewer 93PD** explicitly confirmed: *"Thanks for your detailed rebuttal... Based on these clarifications, I would increase my overall rating and soundness score"*. While we provided equally comprehensive responses and new experiments to **Reviewers fKK3, QC7F, and TNmA**, the discussion period ended before they could respond or update their scores. We believe the major revisions detailed below directly satisfy their stated concerns.

**1\. ADDRESSED: Comparison to Closed Models (GPT-o3) & Scalability**

* **Reviewer Concern:** Do open models lag behind closed proprietary models (like GPT-o3)? Is the method scalable?
* **Our Action:** We scaled our pipeline to a **32B parameter model** (originally 7B).
* **Result:** Our open **32B model now outperforms GPT-o3** on the primary Ranking task (68.05% vs 62.85%) and achieves state-of-the-art results among open models. This proves the method scales effectively and provides reasoning capabilities that closed models struggle with (See **Table 1** in Revision).

**2\. ADDRESSED: Generalization Beyond New Yorker Cartoons**

* **Reviewer Concern:** Does the model overfit to the New Yorker style?
* **Our Action:** We evaluated our model **zero-shot** on two external benchmarks: **YesBut** (NeurIPS 2024\) and **DeepEval** (ACL 2024).
* **Result:** The model demonstrated massive gains over the base model (e.g., **\+31 to \+34%** points on tasks in YesBut, and **\+29 to \+76%** points on tasks in DeepEval), proving that the learned reasoning transfers to other visual humor formats (See **Appendix F**).

**3\. ADDRESSED: Synthetic Data & Human Evaluation Size**

* **Reviewer Concern:** Reliance on synthetic traces and a small human study (N=1 expert).
* **Our Action:** We expanded the user study to **21 diverse participants** to complement the expert evaluation (See **Table 1** in Revision). We also added qualitative comparisons showing how model reasoning aligns with expert strategies (See **Appendix D.4**).

We hope this summary assists your assessment of our work’s contributions to open, interpretable multimodal reasoning.

Sincerely,
The Authors

---

### Meta-Review · Area_Chair_kV2y · 2025-12-08

**Summary:**

This paper presents a three-stage training framework: 1. continued pretraining on humor corpora, supervised 2. fine-tuning with Long chain of thought 3.RL with perception and style rewards, to improve humor understanding for New Yorker style cartoon caption tasks.

The reviewers generally agree that the problem is interesting and acknowledge the clarity of writing.

However, several substantial concerns are consistently raised for the initial version:
1. While the framing around humor is appealing, methodological contribution is perceived as incremental.

2.While improvements over open baselines are shown, performance remains significantly below closed models (e.g., o3).

3. The humor corpus is relatively small and close to NYT cartoons, raising concerns about overfitting.

4. Reliance on synthetic traces and a small human study.

The initial round of reviews reached a consistent consensus leaning negative. While the reviewers acknowledged that the topic is interesting and the paper is clearly written, core concerns were raised regarding novelty, insufficient evaluation, weak performance compared with gpt-o3 and limited human evaluation. The original experimental results did not convincingly demonstrate the necessity or practical benefit of the proposed framework.

In the rebuttal, the authors stated that they conducted substantial additional experiments after submission, including:

Scaling the model from 7B to 32B.

Adding two new external zero-shot benchmarks (YesBut – NeurIPS 2024, DeepEval – ACL 2024).

Expanding human evaluation from 1 expert to 21 diverse participants.

Claiming that the updated results significantly outperform GPT-o3, whereas the original version was substantially weaker.

While such improvements are appreciated, these new experiments majorly alter the empirical findings, conclusions, and claims of the paper, effectively transforming the contribution relative to the originally reviewed version. The newly reported gains (+31 to +34% on YesBut and +29 to +76% on DeepEval) fundamentally change both the result and interpretation compared to the initial manuscript.

However, the new results were not present during evaluation and thus were not scrutinised with the same rigor regarding methodology, reproducibility, dataset details, training cost, ablation relevance, or whether improvements stem from scaling or improved supervision rather than the proposed pipeline itself. The magnitude of revision indicates that the work would require a full new review cycle rather than a conditional acceptance based on late-stage updates.

Therefore, the recommendation for this version of submission is rejection.

**Reviewer Concerns:**

QC7F   GPTo3 outperformed the current method.  While authors scale the model from 7 B to 32 B and outperforms the GPTo3, the newly reported gains (+31 to +34% on YesBut and +29 to +76% on DeepEval) fundamentally change both the result and interpretation compared to the initial manuscript. This significant change should be strictly scrutinised by all reviewers. This is outstanding.

Reviewer fKK3    Model creates a closed loop that train and evaluate on AI, with no validation of whether this aligns with genuine human humour. Authors greatly expanded the user study to 21 diverse participants from only 1 in submitted version. This substance difference has significantly changed any conclusion drawn from initial experiments. Therefore, it requires scrutinise from all reviewers for this brand new version. This concern is outstanding.

93PD  The proposed training framework largely follows a conventional large-model training pipeline. The contribution needs to be validated by strong experiments results. However, initial version does not meet this bar.  This is still outstanding.
 The benchmark coverage is limited. The evaluation focuses only on two datasets, both closely related to NYT-style cartoons.   Authors provided two experiments which are solved.

TNmA  Evaluation of RL rewards (e.g., perception and style judges) still depends on other LLMs using LLM-as-judge. Same as fKK3's concern.
GPTo3 still outperform the initially proposed method in many cases Same as QC7F's concern which is not addressed.

**Reviewer Scores:**

QC7F  may not change from 4
fKK3 may not change from 4
93PD  increase from 2 to 4 since part of concerns are addressed.
TNmA may not change from 4

---

### Decision · Program_Chairs · 2026-01-26

Reject